# Generation of orthotopically functional salivary gland from embryonic stem cells

Junichi Tanaka [1], Miho Ogawa [2,3], Hironori Hojo[4], Yusuke Kawashima[5], Yo Mabuchi [6], Kenji Hata[7], Shiro Nakamura[8], Rika Yasuhara[1], Koki Takamatsu [9], Tarou Irié [1,10], Toshiyuki Fukada[1,5,11], Takayoshi Sakai [12], Tomio Inoue [8], Riko Nishimura [7], Osamu Ohara [5,13], Ichiro Saito[14], Shinsuke Ohba[4], Takashi Tsuji[2,3] & Kenji Mishima [1]

Organoids generated from pluripotent stem cells are used in the development of organ replacement regenerative therapy by recapitulating the process of organogenesis. These processes are strictly regulated by morphogen signalling and transcriptional networks. However, the precise transcription factors involved in the organogenesis of exocrine glands, including salivary glands, remain unknown. Here, we identify a specific combination of two transcription factors (Sox9 and Foxc1) responsible for the differentiation of mouse embryonic stem cell-derived oral ectoderm into the salivary gland rudiment in an organoid culture system. Following orthotopic transplantation into mice whose salivary glands had been removed, the induced salivary gland rudiment not only showed a similar morphology and gene expression profile to those of the embryonic salivary gland rudiment of normal mice but also exhibited characteristics of mature salivary glands, including saliva secretion. This study suggests that exocrine glands can be induced from pluripotent stem cells for organ replacement regenerative therapy.

[1] Division of Pathology, Department of Oral Diagnostic Sciences, School of Dentistry, Showa University, Tokyo 142-8555, Japan. [2] Laboratory for Organ Regeneration, RIKEN Center for Biosystems Dynamics Research (BDR), Kobe, Hyogo 650-0047, Japan. [3] Organ Technologies Inc., Tokyo 101-0048, Japan. [4] Clinical Biotechnology, Center for Disease Biology and Integrative Medicine, The University of Tokyo, Tokyo 113-8655, Japan. [5] Laboratory for Integrative Genomics, RIKEN IMS, Yokohama, Kanagawa 230-0045, Japan. [6] Department of Biochemistry and Biophysics, Graduate School of Health Care Sciences, Tokyo Medical and Dental University, Tokyo 113-8510, Japan. [7] Department of Molecular and Cellular Biochemistry, Osaka University Graduate School of Dentistry, Osaka 565-0871, Japan. [8] Department of Oral Physiology, School of Dentistry, Showa University, Tokyo 142-8555, Japan. [9] Department of Oral and Maxillofacial Surgery, School of Dentistry, Showa University, Tokyo 142-8555, Japan. [10] Division of Anatomical and Cellular Pathology, Department of Pathology, Iwate Medical University, Iwate 028-3694, Japan. [11] Faculty of Pharmaceutical Sciences, Tokushima Bunri University, Tokushima 770-8514, Japan. [12] Department of Oral-Facial Disorders, Osaka University Graduate School of Dentistry, Osaka 565-0871, Japan. [13] Department of Technology Development, Kazusa DNA Research Institute, Chiba 292-0818, Japan. [14] Department of Pathology, Tsurumi University School of Dental Medicine, Yokohama, Kanagawa 230-8501, Japan. These authors contributed equally: Takashi Tsuji, Kenji Mishima. Correspondence and requests for materials should be addressed to K.M. (email: mishima-k@dent.showa-u.ac.jp)

Organogenesis is an essential event according to the body plan during embryogenesis and is a complex process that involves tissue cell–cell interactions, regulations of cell signalling molecules and cell movements. In the embryo, patterning signals indicating body axis and organ-forming fields are strictly controlled by signalling centres according to the embryonic body plan[1,2]. Most organs arise from corresponding placodes via induction by epithelial–mesenchymal interactions in each organ-forming field[3]. Next-generation regenerative therapy consists of organ replacement regenerative therapy, which represents a fundamental approach for treating patients who experience organ dysfunction as the result of disease, injury or ageing[4]. Previous studies provided the proof of concept that fully functional regeneration of ectodermal organs, such as teeth, hair follicles, and salivary and lacrimal glands, could be achieved by reproducing reciprocal epithelial and mesenchymal interactions during embryogenesis by using organ-inductive potential stem cells[5–9]. Organ-inductive stem cells exist in not only embryonal tissues but also adult tissues and regenerating organs. However, several issues remain to be resolved before they can be used for regenerative therapy, such as the cell source for their isolation and the establishment of culture methods for cell expansion and differentiation. Thus, we expected to be developed techniques to regenerate functional organs from pluripotent stem cells (PSCs), such as embryonic stem cells (ESCs) and induced pluripotent stem cells (iPS cells)[10].

PSCs can be induced to differentiate into various somatic cell lineages that mimic the patterning and positioning signals during embryogenesis[11,12]. Several groups have generated neuroectoderm, such as pituitary, optic cup and brain, as well as various organs, including thyroid, intestine, liver, and kidney, generated via the recapitulation of complex patterning signals during embryogenesis and self-formation of PSCs in three-dimensional (3D) organoid cultures[13–18]. Recently, functional integumentary organ system, including skin appendages, was also generated through the reproduction of a skin-forming field by using an in vivo transplantation method[19]. These studies have deepened our understanding of organogenesis in developmental biology and have made a break-through in organ regeneration for use in next-generation organ-regenerative therapy and drug screenings by using partially mimicking organ functions but not specific somatic lineage cells. However, these organoids are still mini-organs, which express partial organ functions and are expected to generate recapitulated organ primordia, which can develop sufficient organ size and then express their functions in vivo from PSCs in 3D stem cell culture[1].

Salivary glands are exocrine glands composed of several lineages, including the ductal, acinar, and basal/myoepithelial cell types. They play essential roles in oral health, including the digestion of starch, swallowing, and the maintenance of teeth through the production of saliva[20]. Salivary glands also arise from their rudiment through a thickening of the primitive epithelium to form a placode in an organ-forming oral field, and their subsequent development by branching morphogenesis depends on epithelial–mesenchymal interactions[21,22]. Salivary gland hypofunction due to radiation therapy for head and neck cancer or Sjogren's syndrome can cause xerostomia, the sensation of a dry mouth[23]. Current therapies for xerostomia involve the administration of artificial saliva substitutes, sialagogues and parasympathomimetic drugs[24]. There have been a few attempts to derive salivary gland cells from PSCs[25,26]. However, functional salivary glands derived from PSCs have not been developed to date. To generate 3D salivary gland tissue from mouse ESCs, it remains unclear which factors define the fate of the primitive oral epithelium (OE). Thus, it is expected that a therapeutic treatment

will be required for the restoration of salivary gland function as an organ replacement therapy.

Here, we successfully regenerated the orthotopically functional salivary gland by using the transplantation of an induced salivary gland primordium (iSG) from mouse ESCs. We identified Sox9 and Foxc1 as critical genes for organ-inductive signals, as they are involved in the commitment of the primitive OE to salivary gland rudiment in the self-organized ESCs. The iSGs secreted saliva after orthotopic transplantation in mice. Our current study provides a proof of concept of a next-generation organ replacement regenerative therapy by using organoid technology.

## Results

**Identification of transcription factors responsible for inducing the salivary gland rudiment from the oral epithelium.** The development of one of the major salivary glands, the sub-mandibular gland (SMG), in mice begins with epithelial thickening of oral mucosa at E11.5, and then, the epithelial invaginates into the underlying mesenchyme (Fig. 1a). SMG development is similar to that of oral-region organs such as the adenohypophysis and teeth, which are mediated through the invagination of OE[27]. Organ-inductive signals including transcriptional factors play essential roles in inducing OE thickening at the initial stage of these organs' development[28]. Therefore, transcription factors related to epithelial thickening at the initial stage of salivary gland development are expected to be useful for inducing salivary gland rudiment from OE differentiated from ESCs. Several transcription factors, including Ascl3, Sox2, and Sox9, are involved in salivary gland development[29–31]. A recent study has shown that FGF10 and Sox9 are required for salivary gland morphogenesis and the expansion of salivary gland epithelial progenitors[31]. Therefore, Sox9 may be important to induce epithelial thickening at the initial stage of salivary gland development, but not sufficient because SMG in Sox9-conditional-knockout mice is arrested at the bud stage[31]. Therefore, it is assumed that there are other factors critical to epithelial thickening at the initial stage. To identify these factors, we investigated the transcription factors that were strongly expressed in the SMG rudiment and neighbouring OE. The mandibles of E12.5 mice were separated into SMG epithelium (bud), invaginating OE connected to the SMG (stalk), and OE distant from the SMG through laser micro-dissection, and the gene expression profiles of these three specimen types were then compared via RNA sequencing (RNA-seq) (Fig. 1b and Supplementary Data 1). A total of 120 genes were commonly up-regulated in bud and stalk, but not in OE (Fig. 1c). There were 22 up-regulated transcription factors, including Sox9 (Fig. 1c and Supplementary Table 1). According to the results of RNA-seq, we picked five genes showing significantly higher expression in SMG bud and stalk compared with OE: EHF, Sox10, Gata3, Cebpb, and Foxc1. Single-copy-gene fluorescence in situ hybridization (FISH) revealed that the gene expression of Sox9 and Foxc1 was strong in OE continuous with stalk and end bud but not OE, suggesting that Foxc1 as well as Sox9 might be an initiation factor to induce placode from OE (Fig. 1d). Sox9 and Foxc1 were significantly up-regulated in the epithelium of the embryonic SMG at E13.5 compared with the OE (Fig. 1e). Additionally, immunofluorescence revealed the presence of Sox9 and Foxc1 in the epithelium of SMG at E13.5, and their expression levels were maintained for a long-term period (from E12.5 to 6 weeks) (Fig. 1d and Supplementary Fig. 1a, b). Sox9 was expressed in most cells of invaginating bud at E13.5 and in epithelial cells of end bud at E16.5. In addition, Sox9 was expressed in acinar cells and intercalated ductal cells at P5 and at 6 weeks. Interestingly, the distribution of Foxc1-positive cells was quite

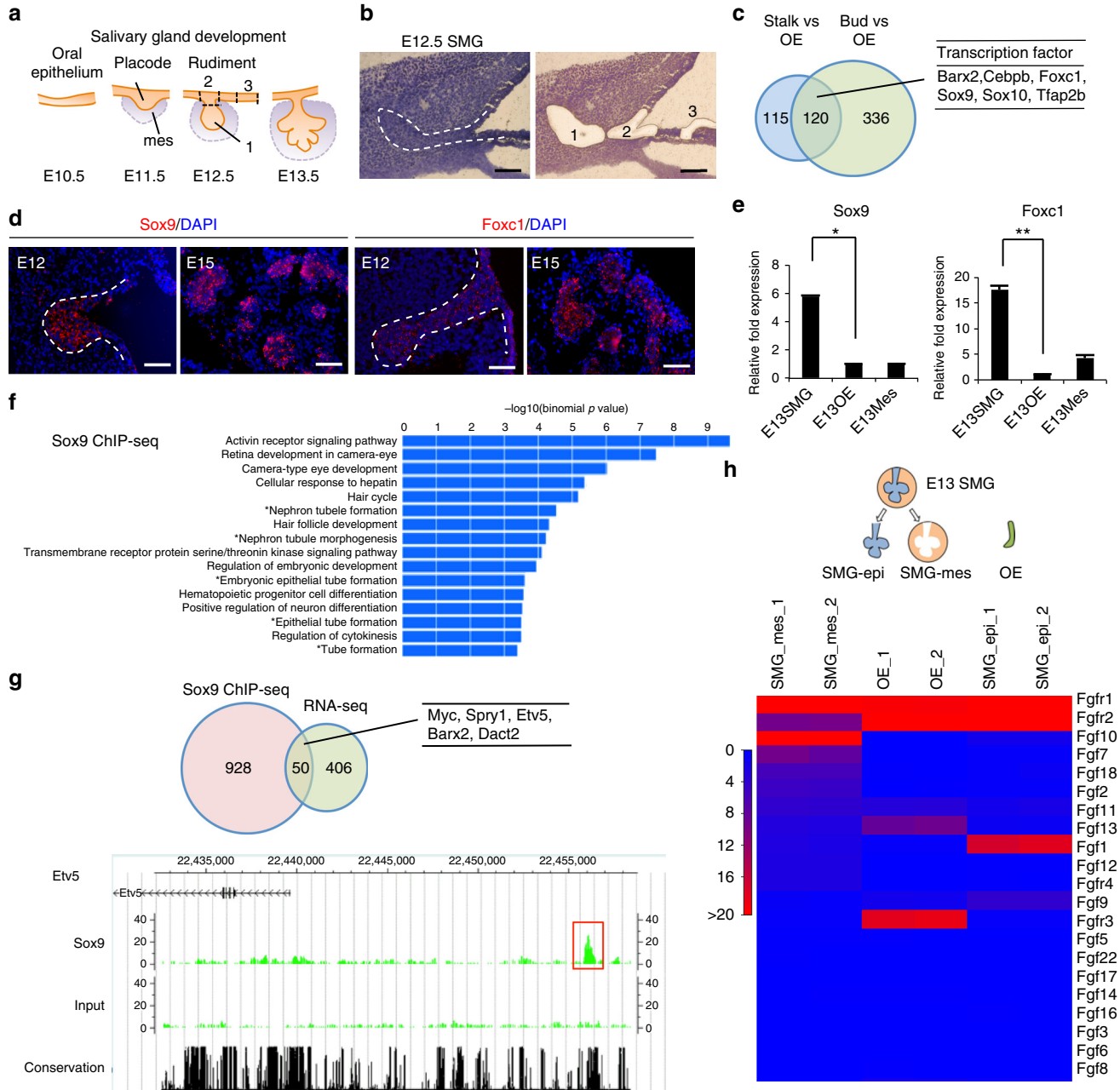

**Fig. 1** Identification of essential factors related to salivary gland development. **a** Schematic representation of salivary gland development. **b** Representative images of submandibular gland (SMG) bud (1), SMG stalk (2), and oral epithelium (OE) distant from SMG (3) before and after laser micro-dissection. The white dashed line indicates SMG epithelium. Scale bars, 50 μm. **c** Venn diagram illustrating the overlapping genes between the genes up-regulated in stalk and bud compared with OE. Up-regulated genes were selected by identifying genes showing fold changes greater than 2.0 and RPKM values higher than 5.0 in stalk and bud. List of common up-regulated transcription factors in stalk and bud. **d** Sox9 and Foxc1 gene expressions were detected in E12.5 and E15.5 SMG by single-copy RNA FISH. The white dashed lines indicate SMG epithelium. Representative images from one out of three embryos are shown. Scale bars, 50 μm **e** Real-time RT-PCR analysis of Sox9 and Foxc1 in the SMG epithelium (SMG-epi), SMG mesenchyme (SMG-mes), and oral epithelium (OE) distant from SMG at E13.5. The results are presented as the mean ± S.D. of triplicate samples and were normalized to GAPDH. Statistical analyses were performed using Student's t-test; *$P = 0.0003$, **$P = 0.00007$. This experiment was replicated three times with similar results. **f** GREAT gene ontology (GO) analysis of Sox9 ChIP-seq results from E13.5 SMGs. **g** Venn diagram illustrating the overlap between the up-regulated genes in E12.5 SMG bud compared with OE in RNA-seq analysis and the neighbor genes of the Sox9 peaks identified from Sox9 ChIP-seq analysis (left). List of overlapping genes involved in salivary gland development (right). CisGenome browser view of Sox9 peaks (red square outline) around the Etv5 region is also shown. Sox9 ChIP-seq (top), an input control (middle), and a conserved region (bottom). The highlighted red box indicates a Sox9 peak region. **h** RNA-seq analysis of the epithelium and mesenchyme of the E13.5 SMG and the OE. Heat map depicting the expression data. Only FGF signalling-related genes are listed in this heat map

similar to that of Sox9-positive ones, and double staining for Sox9 and Foxc1 showed that Sox9-positive cells mostly overlapped with Foxc1-positive cells, at least during E13.5–6 weeks (Supplementary Fig. 1c–e). Foxc1 mediates the BMP signalling required for lacrimal gland development[32], though it remains unclear how Foxc1 is involved in salivary gland development. To examine the functions of Sox9 and Foxc1 in salivary gland development, we next suppressed Sox9 and Foxc1 in organ cultures of E13.5 SMG using two individual siRNAs. The uptake of Cy3-labelled control siRNA in an organ-cultured salivary gland was detected as red fluorescence 1 day after transfection (Supplementary Fig. 2a). The transfection of si-Sox9 or si-Foxc1 inhibited each respective gene's expression compared with the control siRNA (Supplementary Fig. 2b, d). Each gene knockdown inhibited the branching formation, suggesting that Sox9 and Foxc1 are important factors promoting SMG morphogenesis (Supplementary Fig. 2c, e). Interestingly, Sox9 siRNA did not influence Foxc1 gene expression, and Foxc1 siRNA did not influence Sox9 expression either, suggesting that Foxc1 gene expression is not regulated by a Sox9-related pathway.

Sox9 regulates many developmental processes. Sox9 target molecules differ between organs, indicating an organ-specific function of Sox9[33]. To identify genes that are regulated by Sox9 in SMG development, we performed Sox9 chromatin immunoprecipitation sequencing (ChIP-seq) in E13.5 SMG. A total of 760 regions were detected as Sox9-associated genomic regions (Supplementary Data 2), and gene ontology analysis using Genomic Regions Enrichment of Annotations Tool (GREAT) showed that Sox9 peaks were significantly associated with genes related to epithelial tube formation (Fig. 1f). To confirm the integrity of Sox9 ChIP-seq analysis, we performed de novo motif analysis using whole Sox9 peaks. The consensus Sox dimer motif was identified as the top enriched motif (Supplementary Fig. 3a); this motif was enriched in the peak centres (Supplementary Fig. 3b) and mapped to 45% of all Sox9 peaks (Supplementary Fig. 3c). These results suggest that the obtained Sox9 peaks reflect Sox9-mediated biological actions in this context, although the number of peaks was relatively low. To investigate the Sox9-mediated gene-regulatory network specific to salivary gland development, we compared gene lists obtained from Sox9 ChIP-seq studies in salivary glands in this study with previously published pancreatic progenitors[34]. Only 323 genes were shared between the two data sets, supporting the cell-type-distinct Sox9-actions in each organ (Supplementary Fig. 3d and Supplementary Data 3). Fifty genes overlapped between those up-regulated according to RNA-seq in E12.5 SMG and the putative Sox9 target genes located around the top 500 Sox9 peaks in our ChIP-seq analysis (Fig. 1g). In addition, several overlapping genes were involved in these exocrine glands' development, such as Etv5, Myc, Spry1, Barx2, and Dact2 (Fig. 1g and Supplementary Table 2)[35–37]. Therefore, Sox9 seems to be a common factor to induce exocrine gland development, rather than salivary gland specifically. Importantly, consistent with the results of organ culture, Foxc1 may be a non-target gene of Sox9.

After salivary gland placode formation, OE invaginates into the underlying mesenchyme and the epithelium is surrounded by a condensed mesenchyme. Subsequently, salivary gland branching proceeds through epithelial–mesenchymal interactions. Therefore, it is important to recapitulate epithelial–mesenchymal interactions to induce branching formation. It has been well known that several growth factors, such as FGF, can promote SMG development[38–43]. To confirm the expression of growth factors that are secreted from the embryonic SMG-mesenchyme (SMG-mes), we separately isolated SMG-mes, SMG epithelium (SMG-epi), and OE at E13.5, and the gene expression profiles of these three sample types generated through RNA-seq were

compared (Fig. 1h and Supplementary Data 4). As expected, FGF7 and FGF10 were up-regulated in SMG-mes compared with OE and SMG-epi. Consistent with this finding, fibroblast growth factor receptor 2 (FGFR2) was up-regulated in SMG-epi (Fig. 1h). These results indicate that FGF7 and FGF10 secreted from SMG-mes promoted branching and maturation of the salivary gland rudiment.

**Induction of salivary gland rudiment from oral ectoderm by Sox9 and Foxc1.** We next tried to induce oral organ-forming field from self-organized ESCs because the salivary gland rudiment originates as a placode in oral ectoderm (Fig. 2a). Several cytokines, such as BMP4, SB-431542 (inhibitor of TGF-β), LDN-193189 (inhibitor of BMP), and FGF2, play essential roles in inducing non-neural ectoderm, including oral ectoderm, on the outer surface of the ESC aggregates[27,44,45]. We conducted the step-wise induction of oral ectoderm by using SB, BMP4, LDN, and FGF2 each for 2-days, successively, after the embryoid body formation of ESCs for a day (Fig. 2a). A combination of BMP4, SB, LDN, and FGF2 treatment in 3D culture significantly increased the gene expression of Pitx2 (rostral head ectoderm marker) and FGFR2 (receptor of FGF7 and FGF10) compared with their expression in the control on day 8 of differentiation (Fig. 2b, c). In immunofluorescence analyses, the outer surface of the ESC aggregate showed positive fluorescence for the epithelial cell marker pan-cytokeratin (Pan-CK), but not Sox9 (Fig. 2b).

To determine whether Sox9 promotes salivary gland differentiation from oral ectoderm, the outer non-neural ectoderm layers of the aggregates were infected with a recombinant adenovirus encoding Sox9 (Ad-Sox9). We used a recombinant adenovirus encoding β-galactosidase (Ad-β-gal) as a positive control, and successful viral infection was confirmed via β-gal staining at $3 \times 10^6$ pfu (Supplementary Fig. 4a, b). However, Ad-Sox9-infected aggregates cultured for 20 days exhibited only a small amount of Pan-CK-expressing epithelium (Supplementary Fig. 5a, b). Next, on day 8 of differentiation, the outer non-neural ectoderm layers of the infected aggregates were mechanically isolated to maintain an epithelial cell population and were cultured with FGF7 and FGF10 (Fig. 2d). The morphogenetic changes, such as epithelial protrusion of salivary gland buds, were not observed in Ad-Sox9-infected outer layers (Supplementary Table 3). Therefore, we thought Foxc1 could be another candidate to induce differentiation of the primitive OE into salivary gland rudiment. To determine whether the combination of Sox9 and Foxc1 promoted salivary gland differentiation from oral ectoderm, the outer layers of aggregates were infected with Ad-Sox9 and a recombinant adenovirus encoding Foxc1 (Ad-Foxc1) (Supplementary Fig. 5c, d). The infected outer layers were isolated and cultured with FGF7 and FGF10 to recapitulate epithelial–mesenchymal interactions. The dissected regions of ESC aggregates gradually increased in size after 15 days in culture and then showed extensive protrusion from the aggregates towards the outer space, as well as branching, during days 20–28 of differentiation (Fig. 2e–g and Supplementary Fig. 5e, f). The branching structures consisted of Pan-CK-positive epithelial cells, aquaporin 5-positive (AQP5$^+$) acinar-like cells, CK18-positive ductal-like cells, and alpha-smooth muscle actin-positive (α-SMA$^+$) myoepithelial-like cells. AQP5$^+$ cells and α-SMA$^+$ cells were localized at the distal end bud, while CK18$^+$ cells were found on the mesial side (Fig. 2h). Importantly, the branching structure morphologically and immunohistologically mimicked the embryonic SMG (Fig. 2i). These structures arose from a Sox9- and Foxc1-double-positive cell cluster, and their expression was maintained (Supplementary Fig. 5g). On days 23–28 of differentiation, branching morphogenesis continued to expand,

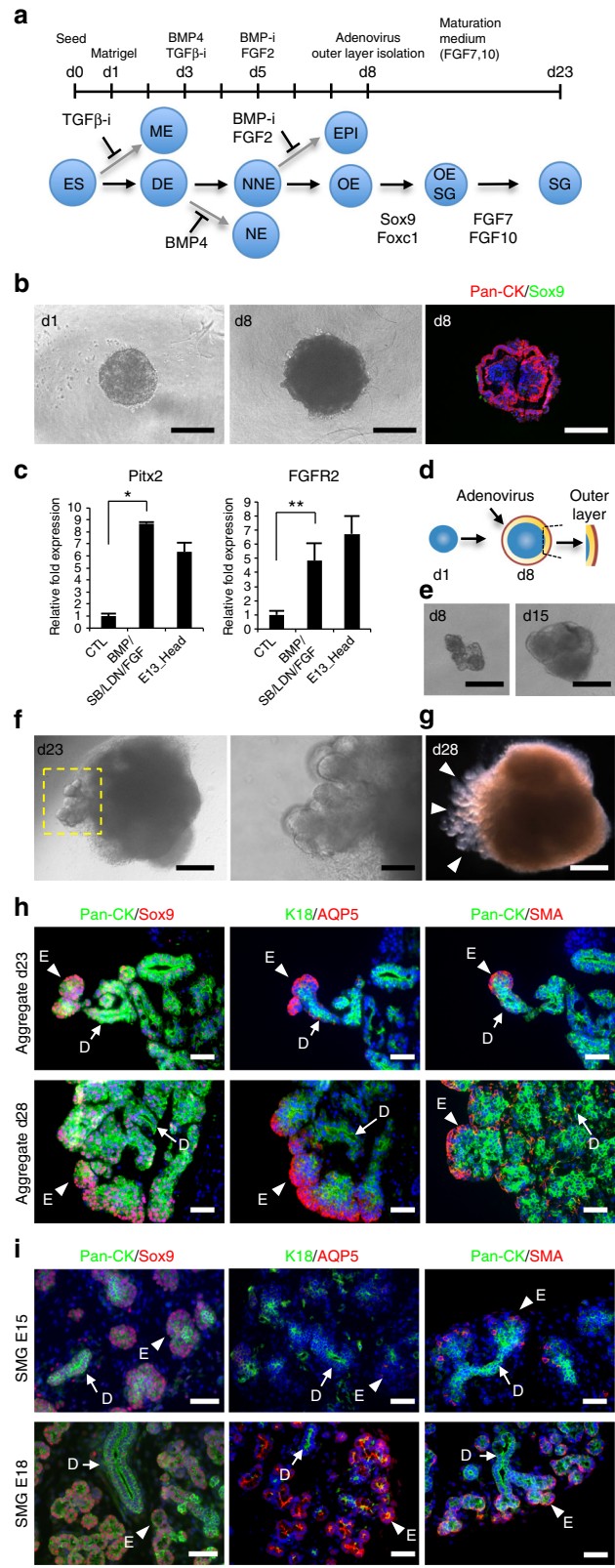

**Fig. 2** Generation of the salivary gland organoid in 3D ESC cultures. **a** Culture protocol (top) and schematic diagram (bottom) of salivary gland differentiation from ESCs. ESCs embryonic stem cells, ME mesendoderm, DE definitive ectoderm, NE neural ectoderm, nne non-neural ectoderm, EPI epidermis, OE oral ectoderm, OE-SG salivary gland placode, SG salivary gland. **b** Phase-contrast representative images of aggregates on day 1 and 8 (left and middle). pan-cytokeratin (Pan-CK, red) and Sox9 (green) were detected via immunofluorescence analysis (right). Scale bars, 300 μm. **c** Real-time RT-PCR analysis of oral ectoderm genes in the aggregates on day 8 with or without growth factors and inhibitors and in the head at E13.5. This experiment was replicated three times with similar results. The results are presented as the mean ± S.D. and were normalized to GAPDH. Statistical analyses were performed using Student's *t*-test; *P = 0.0000009, **P = 0.014. **d, e** The outer layer corresponding to the oral epithelium was resected after infection with Sox9 and Foxc1 adenoviruses and cultured. Schematic representation (**d**) and phase-contrast representative image of the resected outer layer at d8 and the culture at d15. Scale bars, 300 μm (**e**). **f** Phase-contrast representative images showing the morphological changes in the aggregates at d23. The yellow square outline indicates epithelial bud formation (left). Scale bars, 300 μm (left), 50 μm (right). **g** Bright-field view of the aggregate at d28. Arrowheads indicate the epithelial branching structure. Scale bar, 300 μm. **h, i** Immunofluorescence images of the aggregates at d23 and d28 (**h**) and mouse embryonic SMG at E15.5 and E18.5 (**i**). The epithelial marker Pan-CK (green), the ductal marker K18 (green), the acinar cell marker AQP5 (red), and the myoepithelial cell marker α-SMA (red) were detected. Representative images from one out of more than three experiments are shown. Scale bars, 50 μm. Arrowheads indicate an epithelial bud (E). Arrows indicate ducts (D)

The double-positive cells for α-SMA and AQP5 apparently existed as cuboidal cells in end buds of normal E17 SMGs along with SMA-single-positive spindle cells (Supplementary Fig. 6a). In addition, these double-positive cells for α-SMA and AQP5 existed in epithelial bud at d23 (Supplementary Fig. 6b). On d28, α-SMA-single-positive cells appeared in the outer layer of the branching structures (Supplementary Fig. 6c). Moreover, a stem/progenitor cell marker, c-Kit; two lumen markers, Zo-1 and CD133; and a proliferating cell marker, Ki-67, were detected in branching structures (Supplementary Fig. 7a–d). However, several acinar markers, such as PSP and Mist1, were not expressed, suggesting that branching structures derived from ESCs were embryonic salivary glands that had immature acinar cells (Supplementary Fig. 7e, f). These data also suggest that the structure observed on d28 corresponded to normal SMG development between E15 and E18. In addition, microvilli lining the luminal surface and tight junctions between the apical and the basolateral domains of the plasma membrane were identified through transmission electron microscopy (Supplementary Fig. 8). The findings indicate that the structure derived from mouse ESCs was similar to that of the embryonic salivary gland, and this structure will hereafter be referred to as the induced salivary gland primordium (iSG). We have done iSG differentiation experiments in more than 400 ES cell aggregates and repeated them more than 20 times with similar results. To examine whether the combination of two transcription factors and FGF7 and FGF10 were necessary for iSG differentiation, either Ad-Sox9 or Ad-Foxc1 was infected in the presence of various concentrations of FGF7 and FGF10. Crucially, iSG was induced only when all of Ad-Sox9, Ad-Foxc1, FGF7, and FGF10 were present (Supplementary Fig. 9 and Supplementary Table 3).

**In vitro characterization and functional analysis of iSG.** To characterize the iSG, we compared the gene expression profiles of

but the distribution of salivary-specific markers essentially showed no change (Fig. 2h). During normal SMG development, as previously reported, AQP5+ signals were observed in the cytosol and basolateral membrane of the end bud cells of the E18.5 SMG (Fig. 2i), whereas these signals accumulated at the apical membrane surface of end bud cells of the E19 SMG[38].

the iSG, which were dissected from the protruded region of ESC-derived aggregates with those of the embryonic SMG in each developmental stage via RNA-seq (Fig. 3a and Supplementary Data 4). Through hierarchical clustering analysis and principal component analysis (PCA), the iSG gene expression profiles were found to be relatively similar to those observed at E15.5 and E18.5 (Fig. 3b, c). Moreover, the gene expression profiles of the iSG were compared with published datasets for the embryonic and postnatal mouse SMG at any other stages and embryonic other organs[46]. PCA revealed similarity between embryonic SMGs and the iSGs, but not with the E14 lung (GSM2071315) or E15 pancreas (GSM1901172, GSM1901173) (Supplementary Fig. 10a–c). In addition, we compared Pax6 expression, which is a marker of lacrimal gland via real-time RT-PCR[47]. The Pax6 expression level in iSG was similar to that of SMG and significantly lower than those of embryonic and adult lacrimal glands (Supplementary Fig. 10d). These data suggest that the gene expression profile was apparently different in iSG compared to the embryonic stage of other organs, such as lung, pancreas, and lacrimal gland. A heat map of salivary gland-specific gene expression and real-time RT-PCR data indicated that the iSG expressed most of the examined salivary gland markers, including K18, AQP5, α-SMA, and muscarinic receptor-1 (M1) and 3 (M3), but not the pluripotency markers (Fig. 3d, e). Fluid secretion from salivary glands is induced by acetylcholine treatment through the M1 and M3 pathways[48,49]. Treatment with carbachol transiently induced an increase in the intracellular calcium concentration of the iSGs in a dose-dependent manner (Fig. 3f and Supplementary Movie 1, 2). Furthermore, the increase in intracellular calcium observed after acetylcholine treatment was blocked by atropine, which is a muscarinic receptor antagonist (Fig. 3f and Supplementary Movie 3). Thus, the iSG recapitulated the embryonic SMG during E15–18 by showing the morphological, molecular, and functional properties of bona fide salivary glands.

**Functional analyses of orthotopically transplanted iSGs in vivo.** Finally, we investigated whether iSGs that were orthotopically transplanted by using a previously reported method[9], could develop in vivo and show physiological functions, including the correct connection to surrounding tissues and saliva secretion by gustatory stimulation. Briefly, parotid gland-defective mice were prepared, and then the iSGs derived from ESCs (clone G4-2) expressing GFP were transplanted either alone or in combination with E13.5 SMG-derived mesenchymal tissue (SMG-mes), followed by co-culture for 24 h, by using our previously developed orthotopic transplantation method with a guide for duct direction inserted into the iSG (Fig. 4a). The transplanted iSG alone (iSG mesenchyme (−)) successfully developed in vivo with a connection to the recipient parotid gland duct (Fig. 4b–d). Histological analysis using haematoxylin and eosin (H&E) and periodic acid-Schiff (PAS) staining revealed that the developed iSGs had correct structures, including duct and acinar, with GFP fluorescence (Fig. 4c, d). The distributions of salivary gland-specific markers such as AQP5, K18, K5, α-SMA, and NKCC1 detected by immunofluorescence in transplanted iSGs were similar to those in normal salivary glands (Fig. 4e). Most AQP5-positive acinar cells, K18-positive ductal cells, and K5- and SMA-positive basal/myoepithelial cells expressed GFP. AQP5 was localized at the apical membrane of acinar cells, as observed in mature salivary glands (Fig. 4e). iSGs expressed other acinar markers, such as Mist1, PSP, Muc10 (mucous cell marker), and amylase (serous cell marker) (Fig. 4e). The immunofluorescence revealed that the engrafted iSG developed to mature acinar cells containing serous and mucous acinar cells. The iSG transplanted alone included TUBB3-expressing nerve fibres and CD31-expressing vessels,

both of which were derived from GFP-negative recipient cells (Supplementary Fig. 11a). Moreover, the orthotopically transplanted iSGs could connect to the nerve fibres, which were derived from GFP-negative recipient cells, by the detection with anti-neurofilament (Fig. 4f). These results indicate that the iSG could develop according to the process of embryonic salivary gland development in vitro and that orthotopic transplantation of the iSG in vivo could promote its maturation. The iSGs transplanted with mesenchyme also developed in vivo with a connection to the recipient parotid gland duct and showed the expression of salivary gland markers (Fig. 4g–i and Supplementary Fig. 11b–e). Thus, morphologically and immunohistochemically, the transplanted iSG alone as well as iSG transplanted with mesenchyme showed the mature phenotype of salivary gland.

Next, to perform a global and unbiased evaluation of gene expression profiling generated by RNA-seq for the transplanted iSG without mesenchyme, transplanted iSG with mesenchyme, and normal salivary gland of mouse, PCA analysis was applied. PCA analysis revealed that gene expression profile of the transplanted iSG without mesenchyme was quite similar to that of transplanted iSG with mesenchyme and represented those of normal salivary glands between E18 and 6-week-old mice (Fig. 5a and Supplementary Data 4). Therefore, importantly, SMG mesenchyme was not indispensable to the maturation of the transplanted iSG. Next, to examine whether the transplanted iSGs could secrete saliva into the oral cavity in vivo, a sialagogue (pilocarpine) was injected into the major salivary gland-defective mice. The iSGs transplanted alone could not produce enough saliva secretion to measure, while the iSGs transplanted with the mesenchyme produced enough saliva to measure. Therefore, we hereafter evaluated saliva secretion in all major salivary gland defected mice transplanted iSG with mesenchyme (Fig. 5b). Interestingly, saliva secretion after gustatory stimulation with citrate was significantly induced in iSG-engrafted mice compared with water stimulation (Fig. 5c). These results indicate that the engrafted iSGs secreted saliva via the innervations under the control of the central nervous system. We also analysed the protein components secreted from the iSG in saliva (Supplementary Data 5). The fluid secreted from the iSG contained abundant salivary proteins compared to those in previous reports (Supplementary Table 4)[50]. Most protein components secreted in the normal whole saliva were also contained in saliva of iSGs, although several proteins were specifically detected in iSG-derived saliva (Fig. 5d). We expect that these other proteins were derived from contaminated OE, as the filter papers used to collect iSG saliva were directly attached to the oral mucosa. The protein profile of iSG-derived saliva was compared with the gene expression profile of OE[51]. As we expected, 75% other proteins besides saliva proteins, such as keratins and ribosomal proteins, were expressed in the OE (Supplementary Data 6). These results indicate that the orthotopically engrafted iSGs are fully functional in saliva secretion through the reconstruction of the central nervous system comprising afferent and efferent nerves and would be applicable to a future organ replacement regenerative therapy.

## Discussion
In the present study, we successfully demonstrated fully functional iSG by the recapitulation of the embryonic developmental process by the induction of an organ-forming field, transcription factors and maturation factors for the induction of salivary gland rudiment in vitro as an organ model. The iSGs orthotopically secreted saliva, which has salivary secretory proteins, by the reconstruction of neural network in vivo. This study provides the

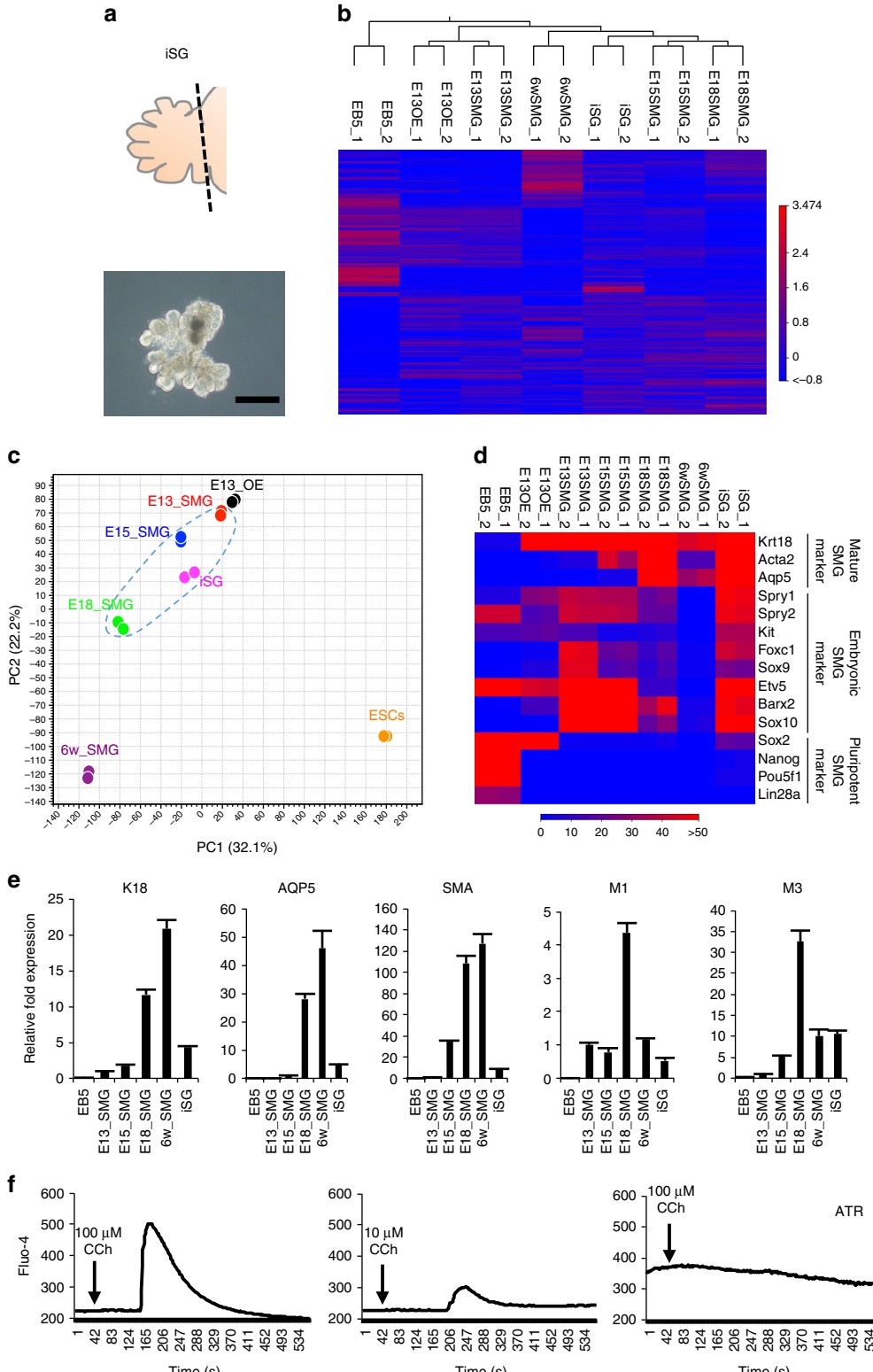

**Fig. 3** In vitro characterization of iSGs. **a** Schematic representation (top) and phase-contrast representative image of the iSG isolated from d23 aggregates (bottom). Scale bar, 300 μm. **b**, **c** A hierarchical cluster analysis (**b**) and principal component analysis (**c**) based on global gene expression examined by RNA-seq. The data are shown for iSGs on d23, mouse ESCs, and SMG at E13.5, E15.5, E18.5, and 6 weeks. The blue circle outline indicates embryonic SMGs. **d** Gene expression levels of pluripotent markers and embryonic and adult salivary gland-specific markers were compared. **e** Real-time RT-PCR verification of the RNA-seq data. Gene expression levels of salivary gland cell lineage markers in ESCs, embryonic and adult SMG, and iSGs were normalized to GAPDH and are presented as the fold change compared with the mean ± S.D. of triplicate samples. This experiment was replicated three times with similar results. **f** iSGs were stimulated with two concentrations of CCh (100 μM, 10 μM) or pretreated with atropine (ATR), followed by 100 μM CCh. Changes in the Fluo-4 fluorescence intensity were recorded. This experiment was replicated three times with similar results

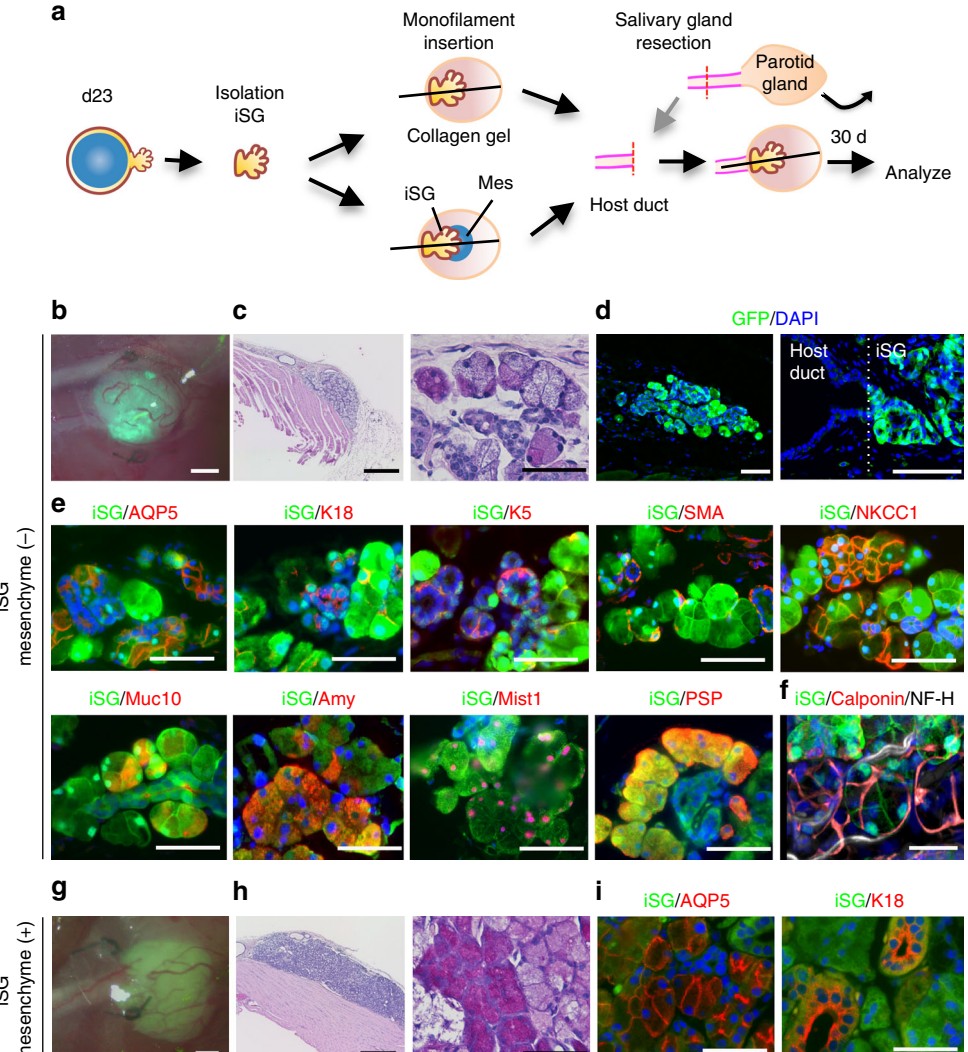

**Fig. 4** Orthotopic transplantation of induced salivary glands. **a** Schematic representation of the protocol for the orthotopic transplantation of iSGs into mice with defective parotid glands. **b** Photograph of GFP-labelled iSG transplants in salivary gland-defective mice on day 30 after transplantation. Scale bar, 500 μm. Representative image from one out of more than 10 times of transplantation are shown. **c** Haematoxylin and eosin (H&E) staining of the transplantation site (left), and a high magnification image of periodic acid-Schiff (PAS) staining (right). Scale bars, 500 μm (left) and 50 μm (right). **d** Fluorescence images of the transplantation site. Most GFP-positive cells formed gland-like structures (left). Higher magnification image showing the duct connection between the host duct and the duct of the iSG (right). Scale bar, 200 μm. **e** Immunofluorescence staining of the salivary gland-specific markers AQP5, K18, K5, α-SMA, NKCC1, Muc10, Amy, Mist1, and PSP. Scale bars, 50 μm. **f** Fluorescent images are projections of 50 μm sections. Merged images of GFP (green), calponin (red), and neurofilament H (NF-H) (white) are shown (right). Scale bars, 50 μm. **g** Photographs of GFP-labelled iSGs transplanted with mesenchymal cells in parotid gland-defective mice on day 30 after transplantation. Representative image from one out of more than 30 times of transplantation are shown. Scale bar, 500 μm. **h** H&E staining (right) and PAS staining (right) of the engrafted iSG with mesenchyme. Scale bars, 500 μm (left), 50 μm (right). **i** Immunofluorescence staining of the engrafted iSG with mesenchymal cells salivary gland-specific markers AQP5 and K18. Scale bars, 50 μm

proof of concept of an organ induction and functional replacement of organoid induced from PSCs for future organ replacement regenerative therapy.

Organogenesis is initiated by the formation of organ rudiments, which is induced via a reaction-diffusion model, a typical Turing model of a specific activator and inhibitor in an organ-forming field according to body patterning[43]. Among regulatory molecules, transcription factors play essential roles in inducing the expression of organ-inductive regulatory molecules[28]. Salivary glands are initiated as a placode in OE at E11.5 and the subsequent organ bud at E12.5 through invagination of the OE along with the underlying mesenchyme[52]. The transcription factors that determine the presumptive sites of SMG development in the

primitive OE have yet to be identified. As previously reported, FGF10-knockout mice show only small initial buds, which degenerate by E12.5[40]. Therefore, the transcription factors regulated by FGF10 are expected to be quite important for early development of SMG. In Sox9-conditional-knockout mice, SMG is arrested at the bud stage, although FGF10-knockout mice show a more severe phenotype than Sox9-conditional-knockout mice[31]. Thus, FGF10 and Sox9 have been expected to be key factors to recapitulate in vivo SMG development. In contrast, FGF7-knockout mice have no SMG phenotype, so this phenotype may be compensated by other pathways[39]. Both FGF7 and FGF10 are produced by mesenchyme around SMG and bind to the same receptor, FGFR2b, expressed on the SMG epithelium, although

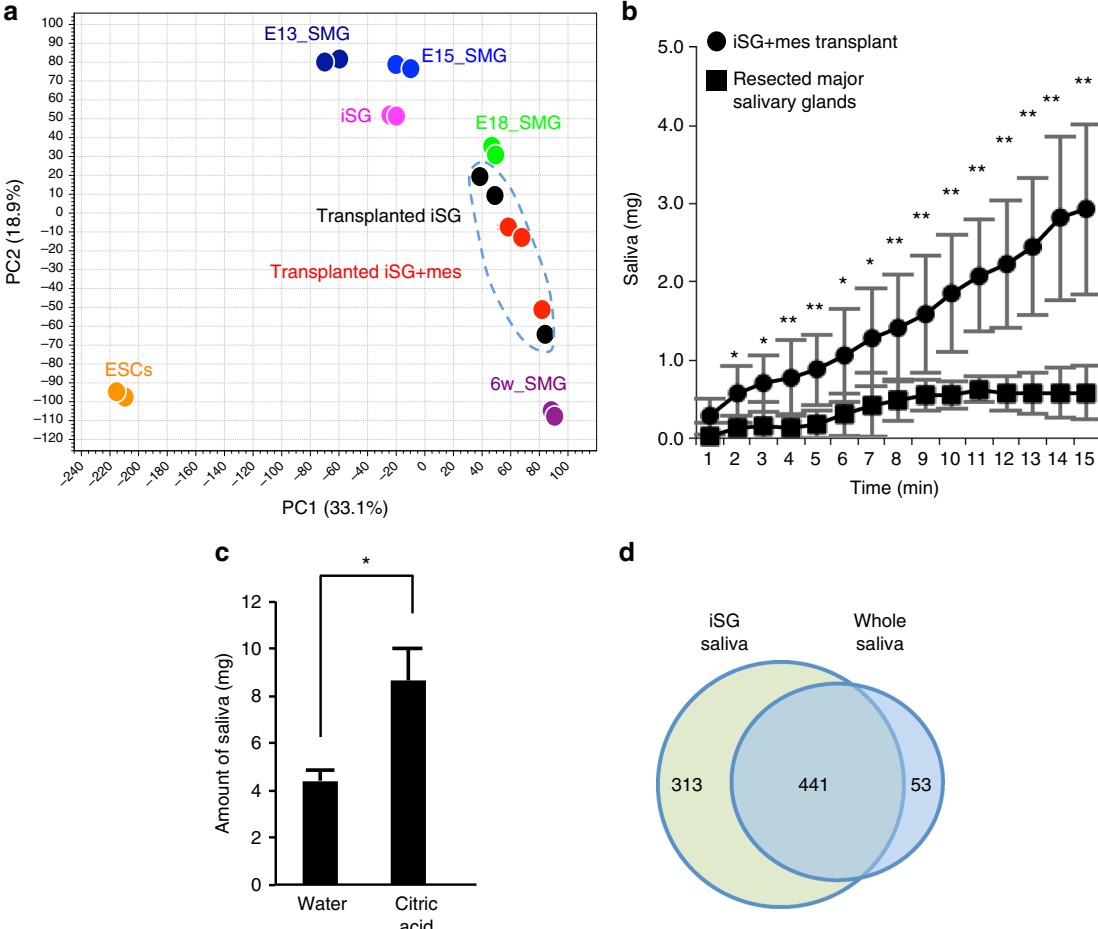

**Fig. 5** Gene expression profile and functional analysis of the transplanted iSG. **a** Principal component analysis based on RNA-seq. The data are shown for iSGs on d23, transplanted iSGs, transplanted iSG with mesenchyme, ESCs, SMG at E13.5, E15.5, E18.5, and 6 weeks. The blue circle outline indicates transplanted iSGs with or without mesenchyme. **b** Time-course of the amount of saliva secretion in major salivary glands-defective mice (square) and combined iSG transplanted major salivary glands-defective mice (circle), every 1 min for 15 min after intraperitoneal injection of pilocarpine; $n = 8$ mice (iSG-transplanted); $n = 4$ mice (major salivary gland defect). $*P < 0.05$, $**P < 0.001$. **c** Assessment of the amount of saliva secretion in combined iSG-transplanted major salivary glands-defective mice for 8 min after water or gustatory (citric acid) stimulation; $n = 4$ mice (each group). $*P = 0.034$. **d** Venn diagram illustrating the proteins between iSG-derived saliva and whole saliva. $n = 3$ mice (each group). The results are presented as the mean ± S.D. Statistical analyses were performed using Student's $t$-test

their functions are different. In SMG explant culture without mesenchyme, FGF7 induces epithelial budding in SMG organ culture without surrounding mesenchyme, while FGF10 induces duct elongation[39]. Although Sox9 plays an important role during salivary gland development, we found that Sox9 could not induce epithelial budding from ESC-derived oral ectoderm. Foxc1 in combination with Sox9 induced the salivary gland bud to undergo epithelial budding and branching morphogenesis from a self-organized ESC-derived oral field in the presence of FGF7 and FGF10. Our results indicate that Foxc1, Sox9, and FGF7 and FGF10 signalling are required for organ-inductive signalling in the oral organ-forming field and induce salivary gland morphogenesis, including duct formation and branching formation. These results provide a strategy, consisting of step-wise induction of organs, including induction of the organ-forming field, the organ rudiment and organ maturation, for initiating organogenesis using specific organ-inductive signalling molecules for various types of organ regeneration.

Organoid studies have demonstrated that various organs, which are composed of multiple cell types and have complex structures, can be regenerated by the recapitulation of the early developmental process of organogenesis from PSCs in vitro[13].

These organoids have somewhat complex organ structures and functions by using complex mini-organs, which will allow them to be applied to drug discovery and basic research on organogenesis. Several of these regenerated organoids, such as pituitary, gastrointestinal tissues, and liver bud, successfully expressed their functions in vivo by the transplantation of a large amount of the organoid[27,53,54]. In the present study, the iSGs expressed specific genes and proteins, including the water channel AQP5, the myoepithelium marker α-SMA, and the muscarinic receptors M1 and M3, that are expressed by normal primordia at E15–18. The iSGs also showed carbachol-induced intracellular calcium up-regulation through muscarinic receptors. Although these findings indicate that the organoids had a complex structure and the potential to replace damaged organs and tissues in clinical applications, it is still unexplored whether the organoids or the regenerated primordia will fully function after orthotopic transplantation, with a sufficient size and correct morphology in vivo[2].

Restoring damaged organ functions and replacing organs with bioengineered organs is expected to be the next-generation of regenerative medicine[2]. Salivary glands play an essential role in oral health, and the reduction of saliva flow causes deterioration of the quality of life[55]. Previous studies have rescued the partial

salivary gland function through the recovery of salivary production by stem cell transplantation or the over-expression of water-channel proteins[56–58]. We have proposed a concept of bioengineered salivary gland organ replacement, but not tissue repair, via the transplantation of a bioengineered germ in a mouse model of salivary defect[9]. In the current study, we showed that the iSG, which connected to the nerve fibres derived from recipient mice, could develop into the correct gland structure following orthotopic transplantation in association with the parotid duct and could produce saliva by muscarinic receptor-mediated stimulation as well as gustatory stimulation by citrate. Here, it is important to emphasize that SMG mesenchyme is not indispensable to induce maturation of the transplanted iSG, because any source of epithelium might be responsible for the addition of SMG mesenchyme. Actually, limb epithelium has been reported to form a branched structure when combined with embryonic SMG mesenchyme[42]. Morphological, immunofluorescence, and PCA analyses revealed that iSGs transplanted without mesenchyme had the mature phenotype of normal salivary gland, suggesting that SMG mesenchyme was not necessary for iSG maturation. However, it remains unclear how SMG mesenchyme contributed to iSG growth. Furthermore, we cannot ignore the possibility of a contribution from non-graft epithelium to the transplanted iSG maturation as well. Recipient-derived innervation may also have influenced the maturation of the transplanted iSG, because parasympathetic innervation can improve salivary gland organogenesis and regeneration[59]. We would like to investigate these issues in future studies.

In conclusion, the current study provides the evidence of the successful replacement of a functional organ through orthotopic transplantation of a self-organized organ rudiment generated from PSCs via step-wise induction, including the organ-forming field and organ-induction signals that recapitulate organogenesis. Further studies investigating organ maturation factors and in vitro culture methods for full functional organ replacement, but not organ rudiment transplantation, will contribute to the future development of next-generation organ replacement regenerative therapy using PSCs.

## Methods

**Animal experiments**. All animal experiments were conducted with the approval of the Institutional Animal Care and Use Committee at Showa University. Female C57BL/6J mice (Clea Japan) were used. C.B-17/lcr-scid/scidJcl mice were purchased from CLEA Japan Inc. All animal studies were conducted with the approval of the Institutional Animal Care and Use Committee at Showa University (permit no. 16036).

**ESC culture for maintenance and differentiation**. ESCs were used for experimentation until passage 40. Mouse ESCs (EB5 (AES0151) and G4-2 (AES0150)) were purchased from RIKEN Bioresource Research Center (BRC) and maintained in G-MEM (Invitrogen) supplemented with 10% knockout serum replacement (Invitrogen), 0.1 mM non-essential amino acids (Invitrogen), 1 mM pyruvate (Sigma), 0.1 mM 2-mercaptoethanol (Wako), and 2000 U/ml LIF (Millipore)[60–62]. For differentiation, ESCs were dissociated with TrypLE Express (Invitrogen), resuspended in differentiation medium, and plated in a volume of 100 μl per well (3000 cells/well) in 96-well low-cell-adhesion U-bottom plates (Nunc). The differentiation medium consisted of Iscove's modified Dulbecco's medium/Ham's F12 1:1 (Invitrogen), 1% chemically defined lipid concentrate (Invitrogen), 450 mM monothioglycerol (Sigma), and 5 mg/ml bovine serum albumin (97% purified through crystallization; Sigma). On day 1, half of the medium was replaced with fresh differentiation medium containing 4% growth factor-reduced Matrigel (BD). On day 3 of differentiation, 25 μl of fresh medium containing 50 ng/ml BMP4 (Sigma) and 5 μM SB-431542 (Stemcell Technologies) was added to each well. On day 5 of differentiation, 25 μl of fresh medium containing 600 nM LDN-193189 (Stemcell Technologies) and 150 ng/ml FGF2 (Peprotech) was added to each well. On day 8 of differentiation, for transcription factor introduction, 70 μl of medium was removed from each well, and adenovirus ($3 \times 10^6$ pfu Ad-Sox9 and $3 \times 10^6$ pfu Ad-Foxc1) was added in 20 μl fresh medium. After a 1-h incubation, the infected aggregates were incubated in dispase (Corning) for 2 min at 37 °C, and the outer layer of aggregates was manually isolated using a 29G syringe needle and then transferred to 24-well low-cell-adhesion plates (Nunc) in maturation medium

containing Advanced DMEM/F12 (Invitrogen), 1% N$_2$ Supplement (Thermo), 1 mM penicillin/streptomycin (Sigma), 1 mM GlutaMAX (Thermo), 1% (v/v) growth factor-reduced Matrigel, 100 ng/ml FGF7 (Peprotech), and 200 ng/ml FGF10 (Peprotech). The maturation medium was changed every other day during floating culture.

**Adenovirus production**. The recombinant adenovirus carrying HA-Sox9, generated via homologous recombination between the expression cosmid cassette and the parental virus genome in HEK293 cells, was provided as a gift by Dr. Riko Nishimura[63]. The recombinant adenovirus carrying HA-Foxc1 was purchased from Applied Biological Materials. Ad-β-gal, which was obtained from RIKEN BRC, was used as a control.

**Single-copy RNA ISH**. Single-copy RNA ISH was performed on frozen sections using a Quantigene ViewRNA in situ hybridization tissue assay (Affymetrix) according to the manufacturer's instructions. A 5-min protease (Protease QF, 1:200) treatment was performed. The probes for mouse Sox9 and Foxc1 (Type 1 probe set) were designed by Affymetrix.

**Ex vivo submandibular gland organ culture**. SMGs were harvested at E13.5 from embryos under a dissecting microscope[64]. For SMG epithelial and mesenchymal tissue separation, SMGs were incubated in dispase for 2 min at room temperature. Using a pair of needles, the epithelial rudiments and mesenchymal tissues were gently separated. The epithelial rudiments were surrounded with mesenchyme on a 0.1-μm pore-sized membrane filter (GE Health Care) floating on DMEM/F12 medium in glass-bottom 50-mm microwell dishes (Iwaki). RNA interference was performed with siRNA. The siRNAs were custom-designed and synthesized by Dharmacon as individual ON-TARGET plus siRNA. The SMGs were transfected with a 500 nM concentration of siGLO Control siRNA, two siRNAs of Sox9, or two siRNAs of Foxc1 using Dharmafect 1 (Dharmacon).

**RNA-seq analysis**. Total RNA was extracted from tissue using the RNeasy Plus Mini Kit (Qiagen) following the manufacturer's instructions. Library preparation from total RNA was performed using the TruSeq Stranded mRNA LT Sample Prep Kit (Illumina) following the manufacturer's instructions. The DNA libraries were sequenced on the Illumina HiSeq 2500 platform. Sequence data were analysed using Seqcutadapt version 1.1, Trimmomatic version 0.32, Tophat version 2.0.14, and Cufflinks version 2.2.1 or the CLC Genomics Workbench. Published RNA-seq data of embryonic SMG, lung, and pancreas were downloaded as fastq files, and compared the overall gene expression profiles with those from the SMGs and iSGs in this study.

**ChIP-seq analysis**. SMG isolated from E13.5 wild-type mice was immediately cross-linked by incubation with 1% formaldehyde (Wako) for 10 min at room temperature, followed by the addition of 0.125 M glycine (Wako) to quench the formaldehyde. A total of 184 SMGs were used per ChIP reaction. Chromatin was sonicated via 10 sessions of 30 pulses (1 s on and 1 s off) at 50% amplitude using a Branson Sonifier 250D (Branson Ultrasonics Corporation). M-280 sheep anti-rabbit IgG Dynabeads (112-03D; Life Technologies) were incubated with the rabbit anti-Sox9 antibody[65]. ChIP-seq libraries were constructed from control samples (input DNA) as well as ChIP DNA using the TruSeq ChIP Sample Prep Kit (Illumina) according to the manufacturer's instructions. Adaptor-modified DNA fragments were enriched via 18 or 22 cycles of PCR. Sequencing was performed using the Illumina HiSeq 2500 platform. DNA-sequence information was aligned to the unmasked mouse genome reference sequence mm9 by bowtie aligner. Peak calling was performed by two-sample analysis on CisGenome package with a FDR cutoff 0.01 compared with the input control[66]. CisGenome browser was used to visualize enrichment of ChIP signals. GREAT gene ontology (GO) was performed using the online GREAT GO program[67]. Each peak category was analysed against whole genome background with assembly mm9. CisGenome package was used to assign a detected peak region to its neighbor gene set[66]. The listed genes include: (a) the nearest gene, which is the gene with the smallest distance from the center of the gene body to the center of the peak region; (b) the nearest upstream gene within 10 M base window; and (c) the nearest downstream gene within 10 M base window. Peaks with identified motifs were obtained by intersection between the peaks and motif-mapped genomic regions by BEDTools-Version-2.16.2. De novo motif analyses were performed using the Gibbs motif sampler provided in the CisGenome package[66]. 100-bp regions surrounding the peak center were extracted from mm9 and used for the analysis. To search for the potential binding protein to the predicted motif, we compared the motif position weight matrices (PWMs) to all known human and mouse motifs in the database. To examine distribution of the identified motif in the peaks, we mapped each PWM back to the whole mouse genome (mm9), comparing to a pre-calculated third-order Markov chain background model; peaks are normalized to 2000-bp window at the peak center. Peak intersection was performed by BEDTools-Version-2.16.2. We compared genes obtained from the salivary gland Sox9 ChIP-seq study (this study) with those from pancreatic progenitors (published data)[34].

**Calcium release analysis**. Fluo-4 (Thermo) was suspended in 0.8% pluronic acid (Thermo) and mixed with Hanks' balanced salt solution (HBSS) without calcium (Invitrogen). iSGs were incubated with the Fluo-4 solution for 30 min at 37 °C. After 30 min, the cells were washed twice with HBSS and incubated for 30 min in HBSS. The changes in fluorescence were captured via image acquisition using a confocal microscope system (Nikon A1R), followed by treatment of the iSGs with 10 or 100 μM of carbachol (Sigma). 0.1 μM Atropine (Tanabe), a muscarinic M1 blocker, was added 15 min before carbachol treatment.

**Immunofluorescence analysis**. Frozen tissue sections were fixed in 4% paraformaldehyde (Wako) and then antigen retrieval was performed by heating at 100 °C in a citrate-buffered solution at pH 6.4. The sections were labelled for 1 h at room temperature with different primary antibodies, including mouse anti-pancytokeratin (1:200 dilution; Cat# C2562, Sigma), rabbit anti-Sox9 (1:2000 dilution; Cat# AB5535, Millipore), mouse anti-Sox9 (1:200 dilution; Cat# AMAb90795, Sigma), rabbit anti-AQP5 (1:200 dilution; Cat# AQP-005, Alomone Labs), mouse anti-K18 (1:50 dilution; Cat# 61028, Progen), rabbit anti-α-SMA (1:200 dilution; Cat# ab5694, abcam), mouse anti-α-SMA (1:200 dilution; Cat# ab7817, abcam), rabbit anti-GFP (1:200 dilution; Cat# 598, MBL), mouse anti-GFP (1:200 dilution; Cat# GTX113617, Genetex), rabbit anti-K5 (1:200 dilution; Cat# ab52635, abcam), rabbit anti-calponin (1:250 dilution; Cat# ab46794, abcam), rabbit anti-NKCC1 (1:200 dilution; Cat# 8351, Cell Signaling), rabbit anti-Mist1 (1:200 dilution; Cat# 148965, Cell Signaling), rabbit anti-Zo-1 (1:200 dilution; Cat# SA241427, Invitrogen), goat anti-c-kit (1:200 dilution; Cat# AF1356, Cell Signaling), rabbit anti-Foxc1 (1:200 dilution; Cat# 8758, Millipore), goat anti-Foxc1 (1:200 dilution; Cat# ab5079, abcam), rabbit anti-CD31 (1:200 dilution; Cat# ab28364, abcam), rabbit anti-TUBB3 (1:200 dilution; Cat# ab18207, abcam), rat anti-NF-H (1:500 dilution; Cat# MAB5448, Millipore), goat anti-Muc10 (1:200 dilution; Cat# EB10617, Everest Biotech), goat anti-PSP (1:200 dilution; Cat# EB10621, Everest Biotech), and mouse anti-Amylase (1:200 dilution; Cat# WH0000276M4, Sigma). The slides were then incubated with a fluorescent secondary antibody (1:200 dilution; Invitrogen). Nuclei were counterstained with DAPI (Nacalai Tesque). Images were acquired using a BZ-9000 fluorescence microscope (Keyence). For staining of NF-H and Calponin, 50 μm frozen sections were labelled over night at 4 °C with primary antibodies. The sections were then incubated with a fluorescent secondary antibody (1:200 dilution) for 2 h at room temperature. Nuclei were counterstained with Hoechst 33342 dye (Life Technologies). Images were acquired using a Laser confocal microscopy (LSM780; Carl Zeiss).

**RNA purification, reverse transcription, and qPCR**. Total RNA was extracted using the RNeasy Plus Mini Kit (Qiagen). Reverse transcription was performed using SuperScript VILO (Thermo). Quantitative PCR was performed using the cDNA samples and a 7500 detection system (Invitrogen). Quantification of the samples was performed according to the threshold cycle using the ΔΔCt method. These experiments were repeated three times. The following primers were employed: Sox9, forward 5′-AAGCCGACTCCCCACATTCCTC-3′, reverse 5′-CGCCCCTCTCGCTTCAGATCAA-3′; Foxc1, forward 5′-CACTCGGTGCGG GAAATGT-3′, reverse 5′-GTGCGGTACAGAGACTGACTG-3′; Pitx2, forward 5′-CGTGTGGACCAACCTTACG-3′, reverse 5′-AAGCCATTCTTGCACAGCTC-3′; CK18, forward 5′-AAGGTGAAGCTTGAGGCAGA-3′, reverse 5′-CTGCAC AGTTTGCATGGAGT-3′; AQP5, forward 5′-GCGCTCAGCAACAACACAAC-3′, reverse 5′-GTGTGACCGACAAGCCAATG-3′; α-SMA, forward 5′-GGAGAAGC CCAGCCAGTCGC-3′, reverse 5′-AGCCGGCCTTACAGAGCCCA-3′; M1, forward 5′-GCCTGTGCCTCAGGATCTAC-3′, reverse 5′-GCTGTACTGGCGCAT CTACC-3′; M3, forward 5′-GGTAGGTGAGTGGCCTGGTA-3′, reverse 5′-GAC ACCTCCAGTGACCCTCT-3′; Pax6, forward; 5′-AGTGAATCAGCTTGGT GGTGTCTT-3′, reverse 5′-TGCAGAATTCGGGAAATGTCGCAC-3′; GAPDH, forward 5′-TGATGACATCAAGAAGGTGGTGAAG-3′, reverse 5′-TCCTTGGA GGCCATGTAGGCCAT-3′. The values presented on the graphs represent the mean ± S.D.

**Electron microscopy**. iSGs were fixed with 2% paraformaldehyde and 2% glutaraldehyde (Sigma) in 0.1 M sodium cacodylate buffer (pH 7.4) and washed three times with 0.2 M sodium cacodylate. The iSGs were subsequently post-fixed with 1% osmium tetroxide for 60 min, dehydrated through an ethanol series, and then embedded in epoxy resin. Ultrathin sections (70 nm thick) were then stained with uranyl acetate and lead citrate and observed under a transmission electron microscope (H-7600; Hitachi).

**Transplantation**. The iSGs were placed into collagen drop and a PGA monofilament (Kono Seisakusho) was inserted to an iSG. The iSG were placed on a cell-culture insert (0.4 μm pore diameter, BD) and incubated at 37 °C for 1 day in maturation medium. To prepare parotid gland-defective mice, the parotid glands of 7-week-old C.B-17/lcr-scid/scidJcl female mice were extracted under deep anaesthesia. The iSG containing a PGA monofilament was placed in the masseter muscle. To create connections between the host parotid duct and the iSG, the PGA monofilament guide was inserted into the host parotid duct, and collagen gel and masseter muscles were fixed using nylon thread (8–0 black nylon 4 mm 1/2 R, Bear Medic Corp)[9].

**Saliva collection and measurement of saliva secretion**. Saliva was collected from the oral cavity using filter paper at 1-min intervals for 15 min after stimulation with an intraperitoneal injection of 300 mg/kg body weight pilocarpine (Wako), and the amount of saliva was calculated. To measure saliva secreted from iSG, SMGs and sublingual glands were extracted by 2 days before measurement (parotid glands resected when transplant iSGs). For measurement of salivary secretion after gustatory stimulation, 5 μl of 0.44 M citric acid (Ken-ei Pharmaceutical Co. Ltd.) or water were placed on the tongue of iSG engrafted and major salivary glands-defective mice. Saliva was measured for 8 min. To perform proteome analysis, saliva of iSG engrafted and major salivary glands-defective mice or normal mice were collected. The iSG saliva was diluted in PBS.

**Proteome analysis**. Whole saliva and iSG-derived saliva proteins were precipitated with acetone. The precipitate was dissolved in phase-transfer surfactant (PTS) buffer (12 mM sodium deoxycholate, 12 mM sodium N-lauroyl sarcosinate in 100 mM Tris–HCl, pH 9.0) through sonication. Protein concentrations were measured using a BCA Protein Assay Kit (Thermo) and adjusted to 100 ng/μl. The dissolved sample was subjected to alkylation with 35 mM iodoacetamide in the dark at room temperature for 30 min after treatment with 10 mM dithiothreitol at room temperature for 30 min. The mixture was then diluted 4-fold with 50 mM ammonium bicarbonate and digested with Lys-C and trypsin for 18 h at 37 °C. An equal volume of ethyl acetate was added to the digested samples, and the mixture was acidified with 0.5% trifluoroacetic acid (final concentration) according to the PTS protocols[68]. The mixture was shaken for 1 min and centrifuged at 15,000×g for 2 min for phase separation, and the aqueous phase was then retrieved. The volume of the digested sample thus recovered was reduced to half or less of the original volume using a centrifugal evaporator for complete removal of ethyl acetate and then desalted with C18-StageTips[69], followed by drying using a centrifugal evaporator. The dried peptides were finally dissolved in 3% acetonitrile and 0.1% formic acid. The peptides were directly injected onto a 75 μm × 15 cm PicoFrit emitter (New Objective, Woburn) packed in-house with 2.7-μm core shell C18 particles (Capcell Core MP; Shiseido), followed by separation using a 150-min gradient at a flow rate of 300 nl/min in an Eksigent ekspert nanoLC 400 HPLC system (Sciex, Framingham). Peptides that were eluted from the column were analysed on a TripleTOF 5600+ mass spectrometer (Sciex) for both shotgun-MS and SWATH-MS analyses[70]. For the shotgun-MS-based experiments, MS1 spectra were collected in the range from 420 to 900 $m/z$ for 250 ms. The top 18 precursor ions with charge states of 2+ to 5+ that exceeded 150 counts/s were selected for fragmentation with rolling collision energy. The dynamic exclusion time was set at 16 s. For the SWATH-MS-based experiments, the mass spectrometer was operated using consecutive data-independent acquisition with 5 $m/z$ increments in the precursor isolation window. Employing an isolation width of 6 $m/z$ (1 $m/z$ for the window overlap), a set of 96 overlapping windows was constructed covering the precursor mass range from 420 to 900 $m/z$. Precursor ions were fragmented for each MS2 experiment using rolling collision energy. All shotgun-MS files were subjected to searches against the mouse UniProt Swiss-Prot database (May 2017 release) using ProteinPilot software v. 4.5 with the Paragon algorithm (Sciex) for protein identification. The protein confidence threshold was a ProteinPilot unused score of 1.3 with at least one peptide showing 95% confidence. The global false discovery rate for both peptides and proteins was lower than 1% in this study. The identified proteins were quantified using SWATH-MS data with PeakView v.2.2 (Sciex). Venn diagram illustrating the proteins whose value are over 25,000 detected between iSG-derived saliva and whole saliva.

**Statistical analyses**. Statistical significance was determined using Student's $t$-test for comparisons of two groups or one-way analysis of variance followed by Tukey's post-hoc test for multiple comparisons. All data were analysed using CLC Genomics Workbench, JMP, or Microsoft Excel software.

## Data availability

The RNA-seq and ChIP-seq data files have been deposited into DDBJ Sequence Read Archive (DRA; https://www.ddbj.nig.ac.jp/dra/index-e.html) with accession numbers DRA007183 and DRA007194, respectively. The proteome analysis data files have been deposited into the Japan ProteOme STandard Repository (JPOST; https://repository.jpostdb.org/) with accession number PXD010541.

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

## Acknowledgements

We thank Dr. H. Yohki (Kyushu University of Nursing and Social Welfare), Ms. Y. Morioka (RIKEN BDR), Ms. Y. Tamai (RIKEN BDR), Ms. M. Shirazaki (RIKEN BDR), and Dr. K. Toyoshima (RIKEN BDR) for technical assistance. This work was supported by JSPS KAKENHI (Grant numbers: 15H05013 to K.M.; 15K20368 and 17K17088 to J.T.) and the MEXT-Supported Program for the Strategic Research Foundation at Private Universities. AMED (Grant number 17bm0404016h0005 to T.T.).

## Author contributions

J.T. designed and performed most of the experiments and analysed the data; M.O. performed the transplantation assay; H.H. and S.O. performed the ChIP sequence analysis; Y.K. and O.O. performed the LC-MS analysis; Y.M., R.Y., K.T., T. Irié, T.F. and T.S. supported the histological analysis; K.H. and R.N. produced the adenoviruses; S.N. and T. Inoue performed the calcium release assay; I.S. supervised and supported the project; T.T and K.M. designed the research plan; and J.T., T.T. and K.M. wrote the paper.

## Additional information

**Competing interests:** T.T. is a director at Organ Technologies Inc. This work was partially performed under the condition of an Invention Agreement between RIKEN and Organ Technologies Inc. The remaining authors declare no competing interests.

