## [Peer Review file · Nature Communications]

Reviewers' comments:

Reviewer #1 (Remarks to the Author):

The authors drive mouse embryonic stem cells to oral ectoderm and form salivary gland-like structures by adding FGF7, FGF10 and overexpressing Sox9 and Foxc1. While the manuscript contains a lot of good data, some of the concepts have been known. The ability to drive ES/iPS cells into salivary gland-like cells has been attempted by others, but this manuscript reveals some of the essential factors to create a functional in vivo SG-like structure (i-SG).

Multiple aspects need to be addressed:

- Page 4: line 16: Organ-inductive stem cells exist only during the embryo state except for hair follicle. What do the authors imply with this? This is in conflict with the literature. This needs correction.
- Page 6: line 11: Salivary gland impairment ... leads to ...salivary gland hypofunction, xerostomia is incorrect and needs to be rephrased.
- Multiple paragraphs need grammatical correction.
- There is no detailed information mentioned on the role of FGF10/7 in organ development. It is not clear why the FGF10-Sox9 relationship is important and so many experiments were needed to highlight this.
- Ext Fig 1: Double staining with acinar or ductal markers is required to outline the 2 cell types. Are Sox9 and Foxc1 present in the same cells?
- ExtData Fig1: a) What does the abbreviation PK stand for? b) pan-CK is mentioned as co-staining but PK in the figure.
- Reference 29 shows Sox9 in the oral epithelium. Why was this factor considered a good candidate gene for the induction of the salivary gland from oral epithelium? More explanation is needed. Why was Foxc1 chosen as a second factor above others?
- Fig 2b. Why do only the outer layers of the ESC aggregate show pan-CK expression?
- Ext Data 4: b) Where is Foxc1 protein expression? Is it in the Sox9 cells? Evidence of Foxc1 transfection needs to be included.
- Page 16, line16: Why do the authors state that only Sox9 and Foxc1 are necessary for i-SG creation? Ext table 3 shows that AdvSox9 and AdvFoxc1 do not form buds on the ESC aggregates without FGF7 and FGF10.
- Page 13, line 15: Reference 29 and 38 do not focus on foxc1 regulating salivary gland development. Needs correction. What is the role of Foxc1 in salivary glands (transgenic mice, siRNA..)?
- Fig 2h-i: In normal glands SMA cells appear as different cell types from AQP5 ones. Do the outer cells in the aggregates express both AQP5 and SMA? Explain this – a higher power image of the double staining needs to be included. Where is Foxc1 in the aggregates?
- Do the AQP and SMA cells come from double or single transduced cells?
- What are the major cellular differences between iSG with or without mesenchyme? A quantitation of the various cell types can help.
- Why are so many other saliva proteins in iSG saliva (#313) that are not present in whole saliva?

Examples/explanation should be given.

- Can the use of filter papers specifically collect saliva from the (parotid) iSG, SMG or sublingual glands? Can contamination with other glandular saliva occur and influence Fig. 4j results?
- How many times have the Adv experiments for organoid creation been performed? Are the results consistent and repeatable?
- FGF7 and FGF10 signal through the same receptor. Are both necessary or is the same effect + AdvSox9/Foxc1 seen with only one?

Reviewer #2 (Remarks to the Author):

This paper has some nice experiments and some impressive findings, mixed in with some data that just confirm known data about salivary glands. I would like to see more acknowledgement of previous work in the field and less presentation of some results as novel.

Major points:

The rationale for starting with E12.5 mandibles is unclear given that this is at the initial bud stage of gland development at least a day after induction of the SG placode.

The text description of the tissues compared by RNAseq is rather poor. The diagram in Figure 1 indicates that the tissue was SG bud, SG stalk rather than SG and epi near the SG. Even at E12.5 there are very large differences in gene expression in these two parts of the gland. The text and images need to be reworded.

It would be nice to also show a comparison of OE-SMG (stalk) with SMG (bud) as well as the comparison to the OE.

The RNAseq results will certainly come up with some interesting genes for further analysis.

Here the authors focus on Sox9 as potentially being involved in gland initiation and show some functional experiments involving siRNA and organ culture at E13.5 (two days after gland initiation).

The problem with this section is that the phenotype of loss of Sox9 in the gland epithelium has recently been published (Chatzeli et al., 2017) and therefore these experiments just confirm what is already known. We know loss of Sox9 has a major affect on salivary gland branching so the results are completely expected.

Interestingly the RNAseq data highlights Sox10 as showing the greatest difference but was not mentioned.

Although a few targets of Sox9 have been investigated in salivary glands (and other branching organs such as the lacrimal gland, pancreas, prostate and lung where Sox9 plays a key role), the ChIP seq analysis allows a comprehensive analysis. It's a pity two different glandular structures that rely on Sox9 were not compared to really pin point salivary gland specific targets, and the data found have not actually been taken forward to move beyond a list of genes.

The RNAseq analysis of the mesenchyme and epithelium at E13.5 comes up with two mesenchymal Fgfs and their receptor in the epithelium. That these Fgfs and their receptor play important roles in salivary gland development has been known for over 18 years. There is no point going to all the expense of an RNAseq experiment if you are looking for candidates that could be identified after a brief search through the literature. Many of the papers confirming the essential role of Fgf10 in gland initiation are not mentioned therefore giving the impression this is new knowledge.

Many of the results from Figure 1, therefore, are not novel and it is questionable why they are shown.

The experiments outlines in Figure 2 are much clearer. The little branches in the organoids at D23 and D28 do have a salivary gland like morphology with clear distinctions between acini/myoepithelial end buds and ductal regions.

Did the authors investigate whether Fgf10 or Fgf7 alone was sufficient for induction of the buds?

The RNAseq data is compared to that from E10 pancreas and liver. At these stages these organs have not started to branch and therefore the branching developing SGs and iSGs are not very comparable. An E11.5 developing SG would also probably not cluster with the iSGs. For this comparison to mean very much the iSGs should be compared to branching stages of other organs. For example are the iSGs very different from branching lacrimal glands, or branching lungs?

For the grafting experiments the authors use iSGs tissue alone and with E13.5 gland mesenchyme. The results from implantation of iSGs alone are very impressive.

The results from adding mesenchymal gland tissue are less impressive and should be taken with some caution. It has been clearly shown in many previous papers using recombinations that salivary gland mesenchyme is able to induce gland development in non-gland tissue. For example limb epithelium will form a branched structure when combined with embryonic salivary gland mesenchyme (Wells et al., 2013). Any source of epithelium, therefore, might be responsive to the addition of gland mesenchyme.

The iSG-generated glands appear to have a contribution from non-graft epithelium. Is this a contribution from the remaining excretory duct?

I am unclear how the authors control for which gland was secreting the saliva after pilocarpine injection. The methods indicate whole saliva was collected. From the methods it appears that the other major glands were removed 2 days prior to collection. So these mice have no major glands other than the transplanted tissue? Was this the case for all the saliva experiments shown in Fig. 4? Perhaps this could be explained more clearly. Was the low level of secretion in the non-iSH mice from the minor glands?

Minor comments:

The salivary glands are referred to as being comprised of two primary lineages, the inner luminal and outer basal-myoeplithelial layer. This is a massive oversimplification, not taking into account the surrounding mesenchyme, the duct and acini structure. If the epithelial lineage was being referred to a duct verses acini/myoeplithelial cell distinction would be better and a reference should be given.

It would be helpful to have some more details on the figures, rather than relying on the legends. For example a key with the identity of the squares and circles in Fig. 4H.

Some of the methods could do with more detail. For example the methods state that two doses of carbacol were used but doesn't say what they were.

The English could do with some help throughout the manuscript

Reviewer #3 (Remarks to the Author):

The manuscript entitled "Generation of orthotopically functional salivary gland from embryonic stem cells" by Tanaka et al., describes the role of two transcriptional regulators, Sox9 and Foxc1, in driving the differentiation of mouse embryonic stem cell derived oral ectoderm into salivary gland rudiments using an organoid culture model system. Transcriptomic studies performed using RNA-sequencing revealed that the transcriptomic identify of the induced salivary gland primordium (i-SG) was similar to wild type embryonic and 6 week old salivary glands. Moreover, orthotopic transplantation of i-SG into mice demonstrated the ability of the primordium to generate a mature salivary gland. While this is an interesting manuscript detailing a proof of concept, there is an overall lack of mechanistic insight into how Sox9 and Foxc1 might function to drive the differentiation of mouse embryonic stem cells to i-SG. This is somewhat disappointing given the vast amount of transcriptomic datasets, which together with the Sox9 ChIP-seq datasets can easily be leveraged to better define a mechanistic role for Sox9 in this differentiation program.

There are additional issues that should be addressed:

- 1) The authors state that the salivary gland is comprised of two lineages (page 6 line 3-4). This can be construed as a rather simplistic interpretation as the salivary gland is a complex organ consisting of several lineages including the ductal, acinar and basal/myoeplithelial cell types.
- 2) Page 10 line 5, the authors use siRNA to knockdown Sox9 expression levels and demonstrate a defect in branching morphogenesis. It appears that the authors utilized one siRNA to knockdown Sox9 expression levels. An additional siRNA should be used to confirm the results.
- 3) The authors have performed ChIP-seq to identify genomic target regions that are bound by the

transcription factor Sox9. Their experiment identified 760 regions that are bound by Sox9 - this seems like a rather low number. Sox9 ChIP-seq studies in other cellular contexts have shown Sox 9 binding to a larger number of genomic regions (~20,000 plus) see for example the paper in Cell Rep. 2015 Jul 14;12(2):229-43. Details about the ChIP experiments are missing and not readily available and should be included.

4) The authors compare the transcriptomic datasets of the i-SG to datasets generated from E13SMG, E15SMG, E18SMG and 6wSMG. It would be valuable to compare the i-SG data set to additional SMG RNA-seq datasets which have been recently reported (Gluck et. al. BMC Genomics, 2016). Including these datasets, which also include additional time points, will further strengthen the results.

5) The authors orthotopically transplanted the i-SG and reported that the developed i-SG resembled a salivary gland. These observations were based on the presence of different structures including ducts and “acinar containing serous and mucous acinar cells” - page 18 line1. However, there is no evidence to support the observations claiming the presence of serous and mucous acinar cells (Fig.4c,d). This statement can be supporting by performing immunofluorescence (IMF) staining using serous and mucous specific markers such as Amylase1 and Muc10 respectively.

6) While the authors transcriptionally profiled the i-SG together with SMGs at different developmental time points, it would be valuable to perform RNA-seq experiments to profile the orthotopically transplanted gland. This is important considering the concerns associated with growing cells in culture, albeit if only for a short amount of time. Comparing the gene expression profiles between the orthotopically transplanted gland and wild type glands would further strengthen the claims that the orthotopically transplanted gland are similar to a normal gland.

7) There are a large number of grammatical errors that need to be fixed.

Response to Reviewer #1

We have studied your comments carefully and found that you understood the value and significance of our study to this field. We are grateful for your evaluation and valuable suggestions for our manuscript. In the manuscript, all changes have been highlighted with yellow colour. Our specific responses are listed below:

Reviewers' comments:

Reviewer #1 (Remarks to the Author):

The authors drive mouse embryonic stem cells to oral ectoderm and form salivary gland-like structures by adding FGF7, FGF10 and overexpressing Sox9 and Foxc1. While the manuscript contains a lot of good data, some of the concepts have been known. The ability to drive ES/iPS cells into salivary gland-like cells has been attempted by others, but this manuscript reveals some of the essential factors to create a functional in vivo SG-like structure (i-SG).

Multiple aspects need to be addressed:

- *Page 4: line 16: Organ-inductive stem cells exist only during the embryo state except for hair follicle. What do the authors imply with this? This is in conflict with the literature. This needs correction.*

<Answer> Thank you for your suggestion. As the suggested, this description is incorrect. Therefore, we corrected this as follows:

Organ-inductive stem cells exist in not only embryonal tissues but also adult tissues and regenerating organs. However, several issues remain to be resolved before they can be used for regenerative therapy, such as the cell source for their isolation and the establishment of culture methods for cell expansion and differentiation.

The text is now shown in on page 4, line 15 - page 5, line 3 of the revised manuscript.

- *Page 6: line 11: Salivary gland impairment ... leads to ...salivary gland hypofunction, xerostomia is incorrect and needs to be rephrased.*

<Answer> As you mention, xerostomia is a sensation of dry mouth and is not always associated with salivary gland hypofunction. Therefore, we rephrased this as follows:

Salivary gland hypofunction due to radiation therapy for head and neck cancer or Sjogren's syndrome can cause xerostomia, the sensation of a dry mouth.

The text is now shown in on page 7, line 11-13 of the revised manuscript.

- *Multiple paragraphs need grammatical correction.*

<Answer> We appreciate this comment and had the manuscript revised by the Springer-Nature English Editing Service.

• *There is no detailed information mentioned on the role of FGF10/7 in organ development. It is not clear why the FGF10-Sox9 relationship is important and so many experiments were needed to highlight this.*

<Answer> Thank you for your suggestion. SMG in Sox9-conditional-knockout mice is arrested at the bud stage, although FGF10-knockout mice show a more severe phenotype than Sox9-conditional-knockout mice. Crucially, FGF10 and Sox9 are required for salivary gland morphogenesis and the expansion of salivary gland epithelial progenitors (Chatzeli L, Development, 144:2294-2305). Thus, FGF10 and Sox9 have been expected to be key factors to recapitulate in vivo SMG development.

In contrast, FGF7-knockout mice have no SMG phenotype, this phenotype may be compensated by other pathways (Steinberg Z, Development, 132:1223-1234). Both of FGF10 and FGF7 are produced by mesenchyme around SMG and bind to the same receptor, FGFR2b, expressed on the SMG epithelium, although their functions are different. In SMG explant culture without mesenchyme, FGF7 induces epithelial budding in submandibular gland organ culture without surrounding mesenchyme, while FGF10 induces duct elongation (Steinberg Z, Development, 132:1223-1234). Indeed, we successfully induced primitive oral epithelium using this organoid culture system, but the underlying mesenchyme, which is specific to salivary gland, was not induced.

As previously mentioned, FGF10-knockout mice show only small initial buds, which degenerate by E12.5. Therefore, the transcription factors regulated by FGF10 are expected to be quite important for early development of SMG. SMG in Sox9-conditional-knockout mice is arrested at the bud stage, although FGF10-knockout mice show a more severe phenotype than Sox9-conditional-knockout mice. Crucially, FGF10 and Sox9 are required for salivary gland morphogenesis and the expansion of salivary gland epithelial progenitors (Chatzeli L, Development, 144:2294-2305). Thus, FGF10 and Sox9 have been expected to be key factors to recapitulate in vivo SMG development.

The text is now shown in on page 27, line 2-14 of the revised manuscript.

• *Ext Fig 1: Double staining with acinar or ductal markers is required to outline the 2 cell types. Are Sox9 and Foxc1 present in the same cells?*

<Answer> As you mentioned, it is important to characterize the two cell types. Therefore, we have conducted double stainings, including AQP5/Sox9 or Foxc1, K18/Sox9 or Foxc1, and

Foxc1/Sox9. At the embryonic stage, Sox9-positive cells were localized in most cells of invaginating buds at E13.5 and in epithelial cells of the end bud at E16.5. At the postnatal stage, Sox9-positive cells were localized in AQP5-positive acinar cells at P5 and localized in AQP5-positive acinar cells and K18-positive intercalated ductal cells at 6 weeks. On the other hand, at the embryonic stage, Foxc1-positive cells were localized in most cells of invaginating buds at E13.5 and epithelial cells of the end bud at E16.5. At the postnatal stage, Foxc1-positive cells were localized in AQP5-positive acinar cells at P5 and localized in AQP5-positive acinar cells and K18-positive intercalated ductal cells at 6 weeks. Thus, the distribution of Sox9-positive cells was quite similar to that of Foxc1. Staining for Sox9 and Foxc1 double-positivity showed that Sox9-positive cells mostly overlapped with Foxc1-positive cells at E13.5, E16.5, P5, and 6 weeks.

This data is now shown in Supplementary Fig. 1b, c, e-f and included in on page 11, line 7-14 of the revised manuscript.

• Ext Data Fig1: a) What does the abbreviation PK stand for? b) pan-CK is mentioned as co-staining but PK in the figure.

<Answer> I apologise for the confusion. Actually, both “PK” and “pan-CK” stand for “pan-cytokeratin”. We use “Pan-CK” throughout the revised manuscript.

• Reference 29 shows Sox9 in the oral epithelium. Why was this factor considered a good candidate gene for the induction of the salivary gland from oral epithelium? More explanation is needed. Why was Foxc1 chosen as a second factor above others?

<Answer> Actually, it has been reported that several transcription factors, including Ascl3, Sox2, and Sox9, are involved in salivary gland development and regeneration (Arany S, Developmental Biology, 2011 353; 186-193, Emmerrson E, EMBO Mol Med. 2018; 10: e8051. Chatzeli L Development. 2017 144:2294-2305). Among these factors, Sox9 was expected to be one of master genes for salivary gland development because SMG in Sox9-conditional-knockout mice is arrested at the bud stage (Chatzeli L Development. 2017 144:2294-2305).

As shown in reference 29, Sox9 expression is localized at the placode (E11.0 and E11.5), which invaginates into the lower mesenchyme, and its expression is maintained throughout salivary gland development. In addition, Sox9 ChIP-seq showed the possibility that the expression of many important factors, including Spry1, Etv5, and Brx2, which are associated with salivary gland development, was directly regulated by Sox9. Thus, Sox9 seems to be one of master regulatory genes of salivary gland development.

Therefore, we expected that Sox9 could induce salivary gland rudiment from ES cell-derived primitive oral epithelium. Next, according to the results of RNA-seq (Fig 1c, Table 1), we picked five genes showing significantly higher expression in SMG and OE-SMG compared with OE: EHF, Sox10, Gata3, Cebpb, and Foxc1. As shown in Fig 1e, gene expression of Foxc1 was

stronger in oral epithelium continuous with stalk compared with other tissues, suggesting that Foxc1 might be one of the initiation factors that induces placode from oral epithelium. Therefore, we focused on Foxc1 as the other factor with Sox9.

Regarding Sox9:

<Answer> The text is now shown in on page 9, line 15 - page 10, line 3 of the revised manuscript.

Regarding Foxc1:

<Answer> The text is now shown in on page 10, line 13 - page 11, line 3 of the revised manuscript.

• Fig 2b. Why do only the outer layers of the ESC aggregate show pan-CK expression?

<Answer> Thank you for your comments. We would like to cite two papers regarding this (Suga H et al., Nature 480, 57-62, 2011 and Koehler KR et al., Nature 500, 217-221, 2013). Specifically, we used SFEBq culture with some modification to induce oral ectoderm from mouse ES cells. In this case, exogenous BMP4 promotes induction of Pitx2-positive oral ectoderm on the surface of ES cell aggregates, which is positive for Pan-CK, and the inner layer of ES cell aggregates consists of Nanog-positive undifferentiated cells.

The text is now shown in on page 15, line 5-8 of the revised manuscript.

• Ext Data 4: b) Where is Foxc1 protein expression? Is it in the Sox9 cells? Evidence of Foxc1 transfection needs to be included.

<Answer> Indeed, it is an important thing to examine the distributions of Foxc1 and Sox9. To examine the distribution of Foxc1 protein expression, we conducted immunofluorescence analysis for Foxc1 after adenoviral infection to express Foxc1. Foxc1 protein was detected on most cells of the outer layer of ES cell aggregates one day after infection. Furthermore, Sox9 showed a similar distribution to Foxc1. The outer layer cells of the bud structure expressed both Sox9 and Foxc1, suggesting that the bud structure was derived from Sox9- and Foxc1-double-positive cells.

These data are now shown in Supplementary Fig. 5d and are included on page 16, line 12 - page 17, line 1 of the revised manuscript.

• Page 16, line16: Why do the authors state that only Sox9 and Foxc1 are necessary for i-SG creation? Ext table 3 shows that AdvSox9 and AdvFoxc1 do not form buds on the ESC aggregates without FGF7 and FGF10.

<Answer> We apologize for this incorrect sentence. We have corrected this sentence as follows:

Thus, the iSG recapitulated the embryonic SMG during E15-18 by showing the morphological, molecular, and functional properties of bona fide salivary glands.

This text can be found on page 21, line 4-7 of the revised manuscript.

• Page 13, line 15: Reference 29 and 38 do not focus on foxc1 regulating salivary gland development. Needs correction. What is the role of Foxc1 in salivary glands (transgenic mice, siRNA..)?

<Answer> As you mentioned, references 29 and 38 do not focus on Foxc1 regulating salivary gland development. Therefore, reference 29 has been deleted, and new sentences have been added as follows:

Foxc1 mediates the BMP signalling required for lacrimal gland development, though it remains unclear how Foxc1 is involved in salivary gland development.

The text can be found on page 11, line 13-15 of the revised manuscript.

Regarding the role of Foxc1:

<Answer> To date, there has been no report on the function of Foxc1 in salivary gland development. Therefore, we examined the function of Foxc1 in salivary gland development using an ex vivo organ culture system. Specifically, we isolated E13.5 SMGs that had been transfected with two different Foxc1 siRNAs (siFoxc1-A, siFoxc1-D). Both Foxc1 siRNAs significantly inhibited Foxc1 gene expression compared with siGlo (control siRNA). Importantly, Foxc1 siRNAs suppressed branching formation of SMGs, suggesting that Foxc1 regulates salivary gland development. Interestingly, Sox9 siRNA did not influence Foxc1 gene expression, which was consistent with the results of Sox9 ChIP-seq, and Foxc1 siRNA did not influence Sox9 expression, either. Taken together, these findings suggest Foxc1 regulates salivary gland development through a different pathway from the Sox9-mediated pathway, although further analysis will be needed to clarify the Foxc1 function in salivary gland development.

These data are now shown in Supplementary Fig. 2d, e and mentioned on page 12, line 3-9 of the revised manuscript.

• Fig 2h-i: In normal glands SMA cells appear as different cell types from AQP5 ones. Do the outer cells in the aggregates express both AQP5 and SMA? Explain this – a higher power image

of the double staining needs to be included. Where is Foxc1 in the aggregates?

<Answer> This is an interesting point. To examine the exact distribution of SMA- and AQP5- positive cells, double staining for SMA and AQP5 was conducted in normal E17 SMGs. Surprisingly, double-positive cells for SMA and AQP5 apparently existed as cuboidal cells in end buds of normal E17 SMGs along with SMA-single positive spindle cells. In addition, these cells existed in the epithelial bud in d23 iSG. Therefore, it is possible that SMA- and AQP5-double positive cells might differentiate into SMA-single positive cells, providing an important direction for future research. In addition, a recent study in mammary gland showed that embryonic multipotent mammary cells of end buds co-express luminal and basal markers (Nature Cell Biology 20, pages 677–687 (2018)).

These data are now shown in Supplementary Fig. 6 and mentioned on page 18, line 2-7 of the revised manuscript.

• Do the AQP and SMA cells come from double or single transduced cells?

<Answer> To clarify this, we have conducted double staining for Sox9 and Foxc1 in ES cell aggregates one day after infection of Ad-Sox9 and Ad-Foxc1 (d9) and ES cell-derived epithelial buds at d20 and d25. Most outer layer cells of ES cell aggregates showed double-positivity for Sox9 and Foxc1. In addition, most cells of epithelial buds at d20 and d25 expressed Sox9 and Foxc1. Therefore, we think that the epithelial bud was derived from double-positive cells and that AQP- and SMA-positive cells came from double-transduced cells.

These data are now shown in Supplementary Fig. 5g and mentioned on page 17, line 11-13 of the revised manuscript.

• What are the major cellular differences between iSG with or without mesenchyme? A quantitation of the various cell types can help.

<Answer> We appreciate your important comment. As the reviewer mentioned, this is very important point. To clarify this, we compared gene expression profiles between transplanted iSGs without mesenchyme, transplanted iSGs with mesenchyme, and normal salivary glands. PCA analysis revealed that the expression profiles of the transplanted iSGs without mesenchyme were quite similar to those of the transplanted iSGs with mesenchyme and normal salivary glands between E18 and 6-week-old mice. Furthermore, we examined the expression of salivary gland markers in the transplanted iSGs with or without mesenchyme, namely, K18 in ductal cells; K5 in basal and progenitor cells; AQP5, NKCC1 (basolateral membranes), Mist1, and PSP in acinar cells; and smooth muscle actin and calponin in myoepithelial cells. The distributions of these markers detected by immunohistochemical analyses of transplanted iSGs without mesenchyme were similar

to those in transplanted iSGs with mesenchyme. These results revealed that iSG without mesenchyme as well as iSG with mesenchyme essentially showed a mature morphological phenotype of normal salivary glands, while the saliva volume between them was different.

These data are now shown in Fig. 4j and mentioned on page 23, line 9-16 of the revised manuscript.

• *Why are so many other saliva proteins in iSG saliva (#313) that are not present in whole saliva? Examples/explanation should be given.*

<Answer> Actually, filter papers used to collect saliva were directly attached to the oral mucosa. Therefore, we expected that the other proteins that were not saliva proteins were derived from contaminated oral epithelium. To examine this possibility, we compared the proteins of iSG-derived saliva that were not present in whole saliva with the gene expression profile of oral epithelium. As we expected, most other proteins besides saliva proteins, such as keratin, were shared in common with the oral epithelium.

These data are now shown in Supplementary Data 6 and mentioned on page 24, line12 - page 25, line4 of the revised manuscript.

Response Figure 1. The percentage of genes of oral epithelium.in iSG-derived saliva that were not present in whole saliva.

• *Can the use of filter paper specifically collect saliva from the (parotid) iSG, SMG or sublingual glands? Can contamination with other glandular saliva occur and influence Fig. 4j results?*

<Answer> We apologize for the confusion. Actually, all major salivary glands, including sublingual and submandibular glands, were removed before collection of saliva (parotid glands were removed during transplantation of iSGs). Therefore, contamination of other glandular saliva did not occur, although that of minor salivary gland-derived saliva cannot be excluded. However, as shown in Figure 4k, the volume of minor salivary gland-derived saliva was significantly low.

The text can be found on page 23, line 16 -page 24, line 6 of the revised manuscript.

• *How many times have the Adv experiments for organoid creation been performed? Are the results consistent and repeatable?*

<Answer> Thank you for these important comments. We have done Adv experiments on more than four hundred ES cell aggregates and repeated it more than twenty times, with similar results.

The text can be found on page 19, line 3-4 of the revised manuscript.

• *FGF7 and FGF10 signal through the same receptor. Are both necessary or is the same effect + AdvSox9/Foxc1 seen with only one?*

<Answer> As you mentioned, FGF7 and FGF10 bind the same receptor, FGFR2b, with high affinity, although FGF10 also binds FGR1b. However, it is expected that the effects of FGF7 and FGF10 are different because FGF7-null mice shows no SMG phenotype, whereas FGF10-null mice show no SMG (Zhou M, 1998, Nature medicine, Jaskoll, 2005 BMC. Dev. Biol). In addition, FGF10 increases endocytic recycling of FGFR2b, which correlates with higher mitogen and degeneration, whereas FGF7 increases receptor ubiquitination and degradation (Belleudi F, 2007, 8; 1854). We have examined whether various concentrations of FGF7 or FGF10 could induce epithelial buds from ES cell-derived primitive oral epithelium. We found that FGF7 and FGF10 only in combination induced epithelial buds after infection by AdvSox9/Foxc1, suggesting that the effect of FGF7 is different from that of FGF10.

These data are now shown in Supplementary Fig. 9 and Supplementary Table 3 and mentioned on page 19, line 4-9 of the revised manuscript.

We hope that these changes meet with your approval. We greatly appreciate your comments, which provided a helpful perspective on our work.

Response to Reviewer #2

We have studied your comments carefully and found that you understood the value and significance of our study in this field. We are grateful for your evaluation and valuable suggestions for our manuscript. In the manuscript all changes have been highlighted with yellow color. Our specific responses are listed below:

Reviewer #2 (Remarks to the Author):

This paper has some nice experiments and some impressive findings, mixed in with some data that just confirm known data about salivary glands. I would like to see more acknowledgement of previous work in the field and less presentation of some results as novel.

Major points:

The rationale for starting with E12.5 mandibles is unclear given that this is at the initial bud stage of gland development at least a day after induction of the SG placode.

<Answer> As you mentioned, it would be best to generate a gene expression profile of the SG placode at E11.5. However, technically, it was very difficult to exactly identify the SG placode and correctly microdissect it. Therefore, we could not help but start with E12.5 to exactly identify the SMG and SMG-related region.

The text description of the tissues compared by RNAseq is rather poor. The diagram in Figure 1 indicates that the tissue was SG bud, SG stalk rather than SG and epi near the SG. Even at E12.5 there are very large differences in gene expression in these two parts of the gland. The text and images need to be reworded. It would be nice to also show a comparison of OE-SMG (stalk) with SMG (bud) as well as the comparison to the OE.

<Answer> Thank you for your comment. As the reviewer mentioned, the diagram in Figure 1 indicates that the tissue was the SG bud and SG stalk rather than SG and epi near the SG. We have corrected this and reworded the text as follows:

To identify these factors, we investigated the transcription factors that were strongly expressed in the SMG rudiment and neighbouring oral epithelium. The mandibles of E12.5 mice were separated into SMG epithelium (bud), invaginating oral epithelium connected to the SMG (stalk), and oral epithelium (OE) distant from the SMG through laser micro-dissection, and the gene expression profiles of these three specimen types were then compared via RNA sequencing (RNA-seq) (Fig. 1b and Supplementary Data 1).

The text can be found on page 10, line 5-11 of the revised manuscript.

The RNAseq results will certainly come up with some interesting genes for further analysis. Here

the authors focus on Sox9 as potentially being involved in gland initiation and show some functional experiments involving siRNA and organ culture at E13.5 (two days after gland initiation). The problem with this section is that the phenotype of loss of Sox9 in the gland epithelium has recently been published (Chatzeli et al., 2017) and therefore these experiments just confirm what is already known. We know loss of Sox9 has a major affect on salivary gland branching so the results are completely expected.

<Answer> We appreciate this comment. Indeed, the phenotype of loss of Sox9 in the gland epithelium has recently been published (Chatzeli et al., 2017). It was important to clarify whether inhibition of Sox9 gene expression influences Foxc1 gene expression. Therefore, using the E13.5 organ culture model, we compared Foxc1 gene expression after transduction of control siRNA with that after transduction of Sox9 siRNA, by real-time RT PCR. Consistent with Sox9 ChIP seq, inhibition of Sox9 gene expression did not influence Foxc1 gene expression.

These data are now shown in Supplementary Fig. 2b-d and mentioned on page 12, line 3-9 of the revised manuscript.

Interestingly the RNAseq data highlights Sox10 as showing the greatest difference but was not mentioned.

<Answer> Thank you for your suggestion. According to the results of RNA-seq (Fig 1c, Table 1), we picked five genes showing significantly higher expression in SMG and OE-SMG compared with OE: EHF, Sox10, Gata3, Cebpb, and Foxc1. As shown in Fig 1e, the gene expression of Foxc1 was stronger in oral epithelium continuous with the stalk compared with other tissues, suggesting that Foxc1 might be one of the initiation factors that induces placode from oral epithelium. Therefore, we focused on Foxc1 as another factor with Sox9.

The text can be found on page 10, line 13-15 of the revised manuscript.

Although a few targets of Sox9 have been investigated in salivary glands (and other branching organs such as the lacrimal gland, pancreas, prostate and lung where Sox9 plays a key role), the ChIP seq analysis allows a comprehensive analysis. It's a pity two different glandular structures that rely on Sox9 were not compared to really pin point salivary gland specific targets, and the data found have not actually been taken forward to move beyond a list of genes.

<Answer> Indeed, it is important to compare the results of Sox9 ChIP-seq analysis in salivary gland with that in other branching organs that rely on Sox9. Therefore, we compared the gene lists obtained from Sox9 ChIP-seq in the salivary glands in this study and previously published pancreatic progenitors. Only 323 genes were shared between the two data sets, supporting the cell-type-distinct Sox9-actions.

These data are now shown in Supplementary Fig. 3d and mentioned on page 13, line 7-12 of the revised manuscript.

The RNAseq analysis of the mesenchyme and epithelium at E13.5 comes up with two mesenchymal Fgfs and their receptor in the epithelium. That these Fgfs and their receptor play important roles in salivary gland development has been known for over 18 years. There is no point going to all the expense of an RNAseq experiment if you are looking for candidates that could be identified after a brief search through the literature. Many of the papers confirming the essential role of Fgf10 in gland initiation are not mentioned therefore giving the impression this is new knowledge.

<Answer> I apologize that many previous reports regarding Fgf10 were not cited in the manuscript. We have cited them in the revised manuscript.

The text can be found on page 14, line 4-9 of the revised manuscript.

Many of the results from Figure 1, therefore, are not novel and it is questionable why they are shown.

<Answer> As you mentioned, many factors, such as Sox9, have already been reported to be important factors. However, we would emphasize that there has been no report regarding Foxc1 involvement in salivary gland development. Therefore, we believe that Figure 1 gave us quite important information justifying the present study.

The experiments outlines in Figure 2 are much clearer. The little branches in the organoids at D23 and D28 do have a salivary gland like morphology with clear distinctions between acini/myoepithelial end buds and ductal regions. Did the authors investigate whether Fgf10 or Fgf7 alone was sufficient for induction of the buds?

<Answer> We are thankful for these encouraging results. We have examined whether various concentrations of FGF7 or FGF10 could induce epithelial buds from ES cell-derived primitive oral endothelium. We found that FGF7 and FGF10 only in combination could induce epithelial buds after infection of AdvSox9/Foxc1, suggesting that the effect of FGF7 is different from that of FGF10.

These data are now shown in Supplementary Fig. 9 and mentioned on page 19, line 4-9 of the revised manuscript.

The RNAseq data is compared to that from E10 pancreas and liver. At these stages these organs have not started to branch and therefore the branching developing SGs and iSGs are not very

comparable. An E11.5 developing SG would also probably not cluster with the iSGs. For this comparison to mean very much the iSGs should be compared to branching stages of other organs. For example are the iSGs very different from branching lacrimal glands, or branching lungs?

<Answer> We agree with your suggestion. We have compared the gene expression profiles of the iSG with published E14 lung and E15 pancreas as the branching stages of other organ datasets. PCA revealed similarities between embryonic SMGs and the iSGs, but not with the E14 lung or E15 pancreas (Supplementary Fig. 10a-c). To confirm whether iSGs are different from embryonic lacrimal glands, real-time RT-PCR for Pax6 (marker of lacrimal glands) was conducted, because there are no deposited RNA-seq dataset of the embryonic lacrimal gland. The Pax6 expression level in iSG was similar to that of SMG and was significantly lower than those of embryonic and adult lacrimal glands (Supplementary Fig. 10d). These data suggest that gene expression profile of the iSG was different from the embryonic stage of other organs, such as lung, pancreas, and lacrimal gland.

These data are now shown in Supplementary Fig. 10a-d and mentioned on page 20, line 1-11 of the revised manuscript.

For the grafting experiments the authors use iSGs tissue alone and with E13.5 gland mesenchyme. The results from implantation of iSGs alone are very impressive.

The results from adding mesenchymal gland tissue are less impressive and should be taken with some caution. It has been clearly shown in many previous papers using recombinations that salivary gland mesenchyme is able to induce gland development in non-gland tissue. For example limb epithelium will form a branched structure when combined with embryonic salivary gland mesenchyme (Wells et al., 2013). Any source of epithelium, therefore, might be responsive to the addition of gland mesenchyme.

<Answer> As you mentioned, this is a very important point. To clarify this, we examined the expression levels of salivary-specific factors in the in vitro-cultured iSGs in more detail. Specifically, we detected the expression of a proliferating cell marker (Ki-67), a stem/progenitor marker (c-Kit), two luminal structure markers (Zo-1 and CD133), and early differentiation markers of acinar cells (AQP5, Mist1, parotid secretory protein (PSP)) in iSGs and embryonic salivary glands. c-Kit-positive cells were detected in iSGs, suggesting that salivary gland stem/progenitor cells existed in iSGs. Consistent with this, Ki-67-positive proliferating cells were also found in iSGs. CD133- and Zo-1-positive cells were distributed in the luminal structures. AQP5 was expressed in iSGs, while Mist1 and PSP were not expressed in iSGs. Specifically, AQP5 expression was distributed in the cytosol and basolateral membrane of acinar cells but was not expressed in the apical membrane. These data, as well as the results of PCA analysis (Figure 3B, C), suggest that iSGs recapitulated natural SMG development, at least until the embryonic stage.

Next, we compared gene expression profiles between transplanted iSGs without

mesenchyme, transplanted iSGs with mesenchyme, and normal salivary glands from 6-week-old mice. PCA analysis revealed that the expression profiles of the transplanted iSGs without mesenchyme were quite similar to those of the transplanted iSGs with mesenchyme and normal salivary glands between E18 and 6-week-old mice. Furthermore, we examined the expression of salivary gland markers in the transplanted iSGs with or without mesenchyme, namely, K18 in ductal cells; K5 in basal and progenitor cells; AQP5, NKCC1 (basolateral membranes), Mist1, PSP in acinar cells; and smooth muscle actin and calponin in myoepithelial cells. The distributions of these markers detected by immunofluorescence in transplanted iSGs without mesenchyme was similar to those in transplanted iSGs with mesenchyme and normal mature salivary glands.

These results reveal that iSG without mesenchyme as well as iSG with mesenchyme essentially showed mature morphological phenotype of normal salivary glands, while the volume between them was different. We concluded that salivary gland mesenchyme was not involved in salivary gland-specific lineage commitment.

These data are now shown in Figure 4j and Supplementary Data 4 and mentioned on page 23, line 9-16 and on page 30, line 6– page 31, line 2 of the revised manuscript.

The iSG-generated glands appear to have a contribution from non-graft epithelium. Is this a contribution from the remaining excretory duct?

<Answer> Thank you for your comments. As shown in Fig 4d, e, in transplanted iSG, most all cells consisting of AQP5-positive acinar cells, K18-positive ductal cells, K5- and SMA-positive basal/myoepithelial cells expressed GFP. Therefore, there was no evidence that no-graft epithelium cells, including salivary gland stem cells, migrated into the transplanted iSG and generated epithelial cells. However, it might be possible that the remaining excretory duct influences iSG maturation via trophic factors. In addition, recipient-derived innervation may influence maturation of the transplanted iSG because parasympathetic innervation improves salivary gland organogenesis and regeneration (Nedvetsky, PI, Dev Cell. 2014 Aug 25;30(4):449-62, Nature Communications volume 4, Article number: 1494 (2013)). We would like to investigate these issues in future studies.

These data are now shown in Supplementary Figure 7 and mentioned on page 18, line 4-12 of the revised manuscript.

I am unclear how the authors control for which gland was secreting the saliva after pilocarpine injection. The methods indicate whole saliva was collected. From the methods it appears that the other major glands were removed 2 days prior to collection. So these mice have no major glands other than the transplanted tissue? Was this the case for all the saliva experiments shown in Fig. 4? Perhaps this could be explained more clearly. Was the low level of secretion in the non-iSH mice from the minor glands?

<Answer> Actually, all major salivary glands, including sublingual and submandibular glands, were removed before collection saliva (parotid glands were removed during transplantation of iSGs). Therefore, contamination with other glandular saliva did not occur, although that of minor salivary gland-derived saliva was not impossible. As shown in Figure 4k, the level of secretion in the non-iSG mice from the minor glands was significantly low.

The text can be found on page 23, line 16 - page 24, line 6 of the revised manuscript.

Minor comments:

The salivary glands are referred to as being comprised of two primary lineages, the inner luminal and outer basal-myoepithelial layer. This is a massive oversimplification, not taking into account the surrounding mesenchyme, the duct and acini structure. If the epithelial lineage was being referred to a duct verses acini/myoepithelial cell distinction would be better and a reference should be given.

<Answer> According to your suggestion, the sentence in the introduction was revised as follows:

Salivary glands are exocrine glands composed of several lineages, including the ductal, acinar and basal/myoepithelial cell types.

The text can be found on page 7, line 5-6 of the revised manuscript.

It would be helpful to have some more details on the figures, rather than relying on the legends. For example a key with the identity of the squares and circles in Fig. 4H.

<Answer> We appreciate this comment. As the reviewer mentioned, we added the details of experiments to the figures.

Some of the methods could do with more detail. For example the methods state that two doses of carbacol were used but doesn't say what they were.

<Answer> According to your comments, we added the details of the experiments to the Methods section.

The English could do with some help throughout the manuscript

<Answer> We appreciate this comment and had the manuscript revised by the Springer-Nature English Editing Service.

We hope that these changes meet with your approval. We greatly appreciate your comments, which provided a helpful perspective on our work.

Response to the Reviewer #3

We have studied your comments carefully and found that you understood the value and significance of our study to this field. We are grateful for your evaluation and valuable suggestions for our manuscript. In the manuscript, all changes have been highlighted with yellow colour. Our specific responses are listed below:

Reviewer #3 (Remarks to the Author):

The manuscript entitled “Generation of orthotopically functional salivary gland from embryonic stem cells” by Tanaka et al., describes the role of two transcriptional regulators, Sox9 and Foxc1, in driving the differentiation of mouse embryonic stem cell derived oral ectoderm into salivary gland rudiments using an organoid culture model system. Transcriptomic studies performed using RNA-sequencing revealed that the transcriptomic identify of the induced salivary gland primordium (i-SG) was similar to wild type embryonic and 6 week old salivary glands. Moreover, orthotopic transplantation of i-SG into mice demonstrated the ability of the primordium to generate a mature salivary gland. While this is an interesting manuscript detailing a proof of concept, there is an overall lack of mechanistic insight into how Sox9 and Foxc1 might function to drive the differentiation of mouse embryonic stem cells to i-SG. This is somewhat disappointing given the vast amount of transcriptomic datasets, which together with the Sox9 ChIP-seq datasets can easily be leveraged to better define a mechanistic role for Sox9 in this differentiation program.

There are additional issues that should be addressed:

1) The authors state that the salivary gland is comprised of two lineages (page 6 line 3-4). This can be construed as a rather simplistic interpretation as the salivary gland is a complex organ consisting of several lineages including the ductal, acinar and basal/myoepithelial cell types.

<Answer> According to your suggestion, the sentence in the Introduction was revised.

The text can be found on page 7, line 5-6 of the revised manuscript.

2) Page 10 line 5, the authors use siRNA to knockdown Sox9 expression levels and demonstrate a defect in branching morphogenesis. It appears that the authors utilized one siRNA to knockdown Sox9 expression levels. An additional siRNA should be used to confirm the results.

<Answer> Thanks to the reviewer for this constructive comment. In the previous version of the article, we used Dharmacon ON-TARGET SMARTpool duplexes, which are provided as a pooled mixture of four siRNAs. To confirm the results, we used two individual ON-TARGET plus siRNAs, which consisted of individual sense-strand sequences. Both sequences of si-Sox9 also inhibited expression of Sox9 and branching morphogenesis.

These data are now shown in Supplementary Figure 2b-d and mentioned on page 12, line 3-9 of the revised manuscript.

3) The authors have performed ChIP-seq to identify genomic target regions that are bound by the transcription factor Sox9. Their experiment identified 760 regions that are bound by Sox9 - this seems like a rather low number. Sox9 ChIP-seq studies in other cellular contexts have shown Sox 9 binding to a larger number of genomic regions (~20,000 plus) see for example the paper in Cell Rep. 2015 Jul 14;12(2):229-43. Details about the ChIP experiments are missing and not readily available and should be included.

<Answer> As you mentioned, 760 regions identified as Sox9-binding ones were fewer than in the previous report. The Cell Rep. 2015 Jul 14;12(2):229-4 mentioned by the reviewer was co-authored by two of the co-authors of the present study (Hironori Hojo and Shinsuke Ohba). Therefore, the Sox9 ChIP-seq used here should have been conducted according to the same protocol as that paper. To confirm the integrity of our Sox9 ChIP-seq analysis, we performed de novo motif analysis using whole Sox9 peaks. The consensus Sox dimer motif was identified as the top enriched motif and was enriched in the peak centres and mapped to 45% of all Sox9 peaks, suggesting that, although the number of peaks was relatively low, the obtained Sox9 peaks reflected Sox9-mediated biological actions in this context.

These data are now shown in Supplementary Fig 3a-c and mentioned on page 13, line 1-7 of the revised manuscript.

4) The authors compare the transcriptomic datasets of the i-SG to datasets generated from E13SMG, E15SMG, E18SMG and 6wSMG. It would be valuable to compare the i-SG data set to additional SMG RNA-seq datasets which have been recently reported (Gluck et. al. BMC Genomics, 2016). Including these datasets, which also include additional time points, will further strengthen the results.

<Answer> According your helpful suggestion, we compared the gene expression profiles of the iSG with additional published SMG RNA-seq datasets (Gluck et. al. BMC Genomics, 2016). PCA revealed that the published SMG RNA-seq datasets (E14, E16, P5, 4ws) and our embryonic SMG dataset (E13, E15, E18, 6w) had similar clusters at respective developmental stages of SMG, and the profiles of iSGs showed similar components to E15, E16, and E18 SMG..

These data are now shown in Supplementary Fig 10a and mentioned on page 20, line 1-5 of the revised manuscript.

5) The authors orthotopically transplanted the i-SG and reported that the developed i-SG

resembled a salivary gland. These observations were based on the presence of different structures including ducts and “acinar containing serous and mucous acinar cells” - page 18 line1. However, there is no evidence to support the observations claiming the presence of serous and mucous acinar cells (Fig.4c,d). This statement can be supporting by performing immunofluorescence (IMF) staining using serous and mucous specific markers such as Amylase1 and Muc10 respectively.

<Answer> We thank the reviewer for this helpful suggestion. According to the reviewer’s suggestion, we performed immunofluorescence for Muc10 (mucinous cell maker) and amylase (serous cell marker) in transplanted iSGs. The immunofluorescence revealed that the transplanted iSG contained serous (GFP- and Muc10-double positive cells) and mucous acinar cells (GFP- and amylase-double positive cells).

These data are now shown in Fig 4e and mentioned on page 22, line 10-13 of the revised manuscript.

6) While the authors transcriptionally profiled the i-SG together with SMGs at different developmental time points, it would be valuable to perform RNA-seq experiments to profile the orthotopically transplanted gland. This is important considering the concerns associated with growing cells in culture, albeit if only for a short amount of time. Comparing the gene expression profiles between the orthotopically transplanted gland and wild type glands would further strengthen the claims that the orthotopically transplanted gland are similar to a normal gland.

<Answer> We appreciate your good suggestion. As the reviewer mentioned, this is a very important point. To clarify this issue, we compared gene expression profiles between transplanted iSGs without mesenchyme, transplanted iSGs with mesenchyme, and normal salivary glands. PCA analysis revealed that the expression profile of the transplanted iSGs without mesenchyme was quite similar to those of the transplanted iSGs with mesenchyme and normal salivary glands between E18 and 6-week-old mice. These results reveal that iSG without mesenchyme as well as iSG with mesenchyme essentially showed the mature morphological phenotype of normal salivary glands.

These data are now shown in Fig 4j and mentioned on page 23, line 9-15 of the revised manuscript.

7) There are a large number of grammatical errors that need to be fixed.

<Answer> We appreciate this comment and had the manuscript revised by the Springer-Nature English Editing Service.

We hope that these changes meet with your approval. We greatly appreciate your comments, which

provided a helpful perspective on our work.

REVIEWERS' COMMENTS:

Reviewer #1 (Remarks to the Author):

Address all the questions.

Reviewer #2 (Remarks to the Author):

The authors have made a good attempt to address my original concerns. They have added more experiments, clarified the text and cited some more original papers. The paper is now significantly improved.

Reviewer #3 (Remarks to the Author):

The authors have adequately addressed my concerns.